# Colonization with extended spectrum beta-lactamase and carbapenemases producing *Enterobacteriaceae* among hospitalized patients at the global level: A systematic review and meta-analysis

Dessie Abera[1,2]*, Ayinalem Alemu[3,4], Adane Mihret[2,5], Abel Abera Negash[2,5], Woldaregay Erku Abegaz[2], Ken Cadwell[6,7]

1 Department of Medical Laboratory Sciences, College of Health Sciences, Addis Ababa University, Addis Ababa, Ethiopia, 2 Department of Microbiology, Immunology and Parasitology, College of Health Sciences, Addis Ababa University, Addis Ababa, Ethiopia, 3 Ethiopian Public Health Institute, Addis Ababa, Ethiopia, 4 Aklilu Lemma Institute of Pathobiology, Addis Ababa University, Addis Ababa, Ethiopoia, 5 Armauer Hansen Research Institute, Addis Ababa, Ethiopia, 6 Department of Microbiology, New York University Grossman School of Medicine, New York, NY, United States of America, 7 Department of Medicine, Division of Gastroenterology and Hepatology, New York University Langone Health, New York, NY, United States of America

* dessabera@gmail.com

## Abstract

### Background

Gut commensal bacteria can mediate resistance against pathogenic bacteria. However, exposure to antibiotics and hospitalization may facilitate the emergence of multidrug resistant bacteria. We aimed to conduct a systematic review and meta-analysis to provide comprehensive evidence about colonization rate of extended spectrum beta-lactamase and carbapenemases producing *Enterobacteriaceae*.

### Method

We used PubMed, Google Scholar and Web of Science data bases to search studies from January 1, 2016 to August10, 2022 about colonization rate of extended spectrum beta-lactamase and carbapenemase producing *Enterobacteriaceae*. Data were extracted from eligible studies and analyzed using Stata version 16 software. The quality of included studies was assessed using the Joanna Briggs Institute Critical Appraisal tools, and publication bias was assessed using funnel plot and eggers test.

### Results

We identified 342 studies from the comprehensive data search and data were extracted from 20 studies. The pooled estimate of extended spectrum beta-lactamase and carbapenemase producing *Enterobacteriaceae* were 45.6%(95%CI: 34.11-57-10) and 16.19% (95% CI: 5.46–26.91) respectively. The predominant extended spectrum beta-lactamase

**Data Availability Statement:** All relevant data are within the paper and its supporting information files.

**Funding:** The authors received no specific funding for this work.

**Competing interests:** The authors have read the journal's policy and have the following competing interests: K.C. has received research support and/or consulting fees from Pfizer, Takeda, Pacific Biosciences, Vedanta, Genentech, and Abbvie. The authors would like to declare the following patents/patent applications associated with this research: U.S. patent 10,722,600 and provisional patent 62/935,035 and 63/157,225. This does not alter our adherence to PLOS ONE policies on sharing data and materials.

producers were *E. coli*,32.99% (95% CI: 23.28–42.69) and *K. pneumoniae*, 11.43% (95% CI:7.98–14.89). Prolonged hospitalization was linked to carbapenemase producing *Enterobacteriaceae* colonization with the odds of 14.77 (95% CI: -1.35–30.90) at admission and 45.63 (95% CI: 0.86–92.12) after ≥7 days of admission.

## Conclusion

The pooled estimate of extended spectrum beta-lactamase and carbapenemase producing *Enterobacteriaceae* were high. This indicates the need for strong mitigation strategies to minimize the spread of multidrug-resistant bacteria at the healthcare facilities.

## 1. Introduction

The human gut contains a myriad of commensal bacteria that can mediate resistance [1]. However, exposure to antibiotics may reduce the number and function as well as exert selection pressure on the gut microbiome which may promote the emergence of multidrug-resistant bacteria [2]. In particular, hospitalized patients are at increased risk of developing infections because of their lower immune status and occurrence of multidrug-resistant bacteria (MDRB) [3]. The emergence of MDRB in combination with altered gut microbiome may complicate patients' outcome [4]. Gram-negative bacteria, especially *Enterobacteriaceae* are frequently reported in the hospital settings. These bacteria are well known in producing extended-spectrum beta-lactamases (ESBLs) and carbapenemase. *E. coli* and *K. pneumoniae*, in particular, have increased rapidly at the global level in both hospital and community settings [5,6].

The mechanisms how MDRB emerge in the hospital settings are complicated, but few mechanisms have been proposed. The first way is, resistant bacteria may get into the hospital at the time of admission of patients who already have colonized with resistant strain. Secondly, during prolonged hospitalization, susceptible bacteria may acquire resistant gene through conjugation mechanism or through genetic mutation [7]. There are many factors that can facilitate the colonization rate of MDRB in hospitalized patients: such as low hygiene practice, improper antibiotic usage, prolonged hospitalization, implanted materials, sharing a single room with high carrier patients including door knobs, wheelchairs and sinks have been associated [8–10]. In addition, patients may remain colonized by MDRB after they leave the hospital and become a potential source of resistant gene transmission for the community [11].

Despite the improvement of using antimicrobial stewardship including personal and environment hygiene, colonization with multi drug-resistant bacteria in the hospitalized patients is still a global challenge [12]. Therefore, screening MDRB carriage of patients prior to admission, during admission and before discharge is critical for mitigating the transmission of resistant bacteria from hospital to community and community to hospital settings [13]. Isolation room for MDRB carrier patients is also another recommended mechanism to prevent transmission of resistant bacteria, but lack of resources may limit the widespread use of this approach [14,15]. Many studies have been conducted on colonization of MDRB from multiple sample sources by focusing on a specific geographical location [16–19]. However, there is limited information whether there are factors that determine colonization with extended spectrum beta-lactamase producing *Enterobacteriaceae* (ESBL-PE) and carabapenem resistant *Enterobacteriaceae* (CRE) that are true across distinct geographic regions. Therefore, the

current systematic review and meta-analysis aimed to assess colonization rate and risk factors of ESBL-PE and CRE colonization among hospitalized patients at the global level.

## 2. Methods

### 2.1. Search strategy and study selection

The Preferred Reporting Items for Systematic Reviews and Meta-analysis (PRISMA) was followed to conduct this study [20]. We used PubMed, Google scholar and Web of Science as search strategies. Articles were identified using MeSH terms and keywords of the title and by using Boolean operators (OR, AND, NOT) such as Factors OR Bacteria OR ("Gastrointestinal Microbiome/drug effects"[Mesh] OR "Gastrointestinal /etiology"[Mesh] AND Carriage OR Colonization OR ("Asymptomatic Infections/epidemiology"[Mesh] OR "Asymptomatic Infections/mortality"[Mesh] OR "Asymptomatic Infections/nursing"[Mesh] OR "Asymptomatic Infections/therapy"[Mesh]) AND Resistance OR ("Drug Resistance OR Bacterial/drug effects"[Mesh] OR "Drug Resistance OR "Extended-spectrum betalactamase-producing *Enterobacteriaceae* /carbapenemase-producing *Enterobacteriaceae*"[Mesh] OR "Drug Resistance OR Bacterial/etiology"[Mesh] AND Inpatient OR Hospitalized ORPatient OR("CrossInfection/complications"[Mesh]OR"CrossInfection/diagnosis"[Mesh]OR"CrossInfection/drugtherapy"[Mesh]OR"CrossInfection/epidemiology"[Mesh]OR"CrossInfection/etiology"[Mesh]. All studies starting from January 1, 2016 to August 10, 2022 were assessed, extracted and refined. All citations were exported to Endnote software and duplicates were removed.

### 2.2. Eligibility criteria

Selection citeria checklist was developed by the authors before identifying appropriately published relevant full-text articles.

**Inclusion**: All studies which fulfilled at least the following criteria were incorporated in this systematic review and meta-analysis: Studies from January 2016 to August 10,2022, case-control, cohort and cross-sectional studies with human fecal specimens, hospitalized patients, studies reported ESBL-PE and/ or CRE, with or without MDR of *Enterobacteriaceae*.

**Exclusion:** Case reports, retrospective studies, unrelated articles and specimens other than stool or fecal were excluded.

### PICOS criteria

Participants: Any hospitalized patients
   Intervention: Not applicable
   Comparator: Not applicable
   Outcome: ESBL-PE, CRE, MDR, *E. coli*, *K. pneumoniae*, odds ratio
   Study design: Case-control, cohort and cross-sectional
   Study setting: Any setting across the globe

### 2.3. Data Screening, extraction, and management

Online records from different databases and directory were exported properly to ENDNOTE reference software version 8.2. The records were merged into one folder to identify and remove duplicate articles with the help of ENDNOTE or manual tracing way as there are several possibilities of citation styles per article. Data screening was performed by three reviewers (DA, AA and AAN) who independently screen the titles and abstracts of all relevant articles from literature search databases based on the predefined eligibility criteria.

Data extraction format included principally first author, study design, study setting, publication year, study site in the country, sampling technique, sample size, population characteristics, the age group of study participants, the prevalence of ESBL-PE, diagnostic methods, specimen types, and types of ESBL-PE isolates. In cases of insufficient/ incomplete data, the authors independently reviewed the full-text of the article for further information and clarification. Authors designed a data extraction form adopted from the Preferred Reporting Items for Systematic Reviews and Meta Analyses (PRISMA), 2020 flow diagram. Finally, a total of 20 eligible articles were included in the study as indicated in (Fig 1) [20]. PRISMA 2020 check list

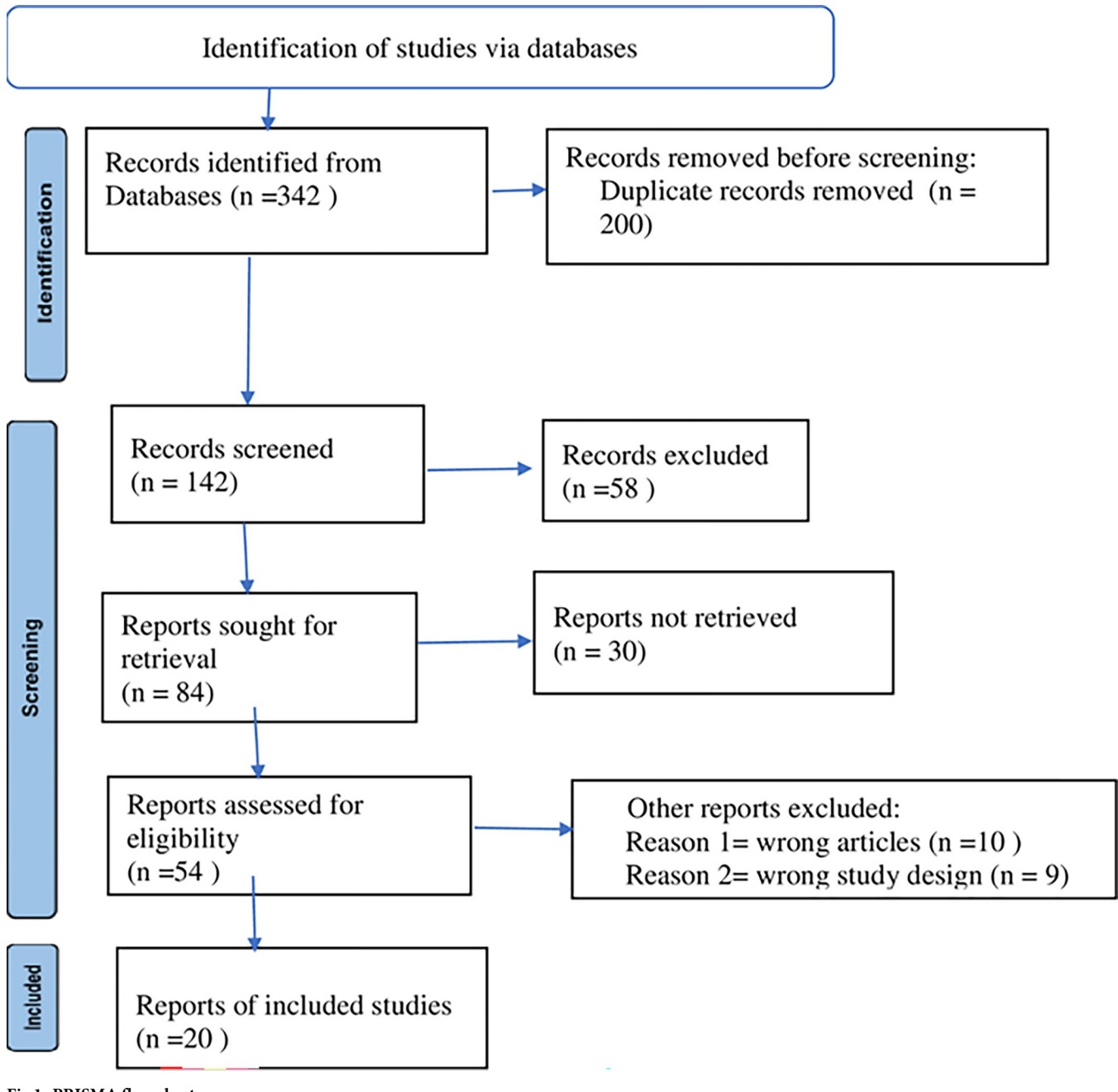

**Fig 1. PRISMA flow chart.**

was used to indicate detail descriptions found as shown in the supplementary file (see S1 Checklist).

## 2.4. Study population

All age groups, gender (male/female) and any ethnic groups live in different countries in most of the world were included in this study.

## 2.5. Outcome

The main outcomes of interest were the prevalence of ESBL-PE colonization, CRE colonization, MDR and risk factors among hospitalized patients from fecal specimens across the globe. *E. coli*, *K. pneumoniae* and comorbidities were also measured.

## 2.6. Quality assessment

Critical appraisal of the studies was performed by assigned reviewers (authors) to ensure the accuracy and consistency of data. The quality of studies was assessed using standard critical appraisal tools prepared by Joanna Briggs Institute (JBI) [21]. The purpose of the appraisal was to assess the methodological quality of studies, the possibility of bias in its design and statistical analysis. The JBI appraisal checklist contains 8, 10 and 11 questions for cross-sectional, case-control and cohort studies respectively. Two independent authors (AA and AAN) were assigned to assess the quality of each study and the discrepancies were solved by taking the average score. Finally, five point (5) was taken as a cut of point, and studies with a score of five and above for "yes" were included in the systematic review and meta-analysis. The total score is calculated by counting the number of "yes" in each row and variables are scored out of 100% and quality score is defined as low if < 60%, medium if 60–80% and high if > 80%. Forty five percent of the studies have high level of quality score as indicated in (Table 1).

## 2.7. Data synthesis, analysis, and reporting

Data were imported from Microsoft Excel 2019 to Stata version16.0MP software for statistical analysis. The pooled estimation of ESBL-PE, CRE, MDR and odds ratio with a 95% confidence interval were performed. Random effect model was used to assess heterogeneity ($I^{2)}$) between studies. According to Higgins *et al* studies: $I^2 \leq 25\%$ is low heterogeneity, $I^2 = 50\%$ is moderate heterogeneity, and $I^2 \geq 75\%$ is high heterogeneity [22]. Moreover, publication bias was also assessed using funnel plot and eggers test [23].

## 2.8. Ethical approval and consent to participate

Since this study was based on data extracted from previously published studies, ethical approval was not applicable.

# 3. Result

## 3.1 Characteristics of included studies in systematic and meta-analysis

From the available scientific database 342 studies were identified. Of this, 200 duplicate studies were removed. One hundered forty two studies were screened by title and abstract. Profound review was made using their titles, study design, outcomes, and other important parameters using checklists and quality assessment tools. The remaining 34 studies were excluded from the analysis based on specimen source, methodological difference and being related to the

**Table 1. Quality assessment of the studies included in the systematic review and meta-analysis.**

| Author | Design | Q1 | Q2 | Q3 | Q4 | Q5 | Q6 | Q7 | Q8 | Q9 | Q10 | Q11 | Scored | 100% | Quality |
|---|---|---|---|---|---|---|---|---|---|---|---|---|---|---|---|
| Desta, *et al* | CS | Yes | Yes | Yes | Yes | No | No | Yes | Yes | | | | 6 | 75 | Medium |
| Baljin, *et al* | C | Yes | Yes | Yes | No | No | No | Yes | Yes | No | Yes | Yes | 7 | 64 | Medium |
| Aklilu, *et al* | CS | No | Yes | Yes | Yes | Yes | Yes | Yes | Yes | | | | 7 | 88 | High |
| Mahamat, *et al* | CC | Yes | Yes | UC | Yes | Yes | No | No | Yes | No | Yes | | 6 | 60 | Medium |
| Abdallah, *et al* | CS | Yes | Yes | Yes | Yes | No | No | Yes | Yes | | | | 6 | 75 | Medium |
| Ouédraogo, *et al* | CC | Yes | Yes | Yes | Yes | Yes | No | No | Yes | No | Yes | | 7 | 70 | Medium |
| Ebrahimi, *et al* | CS | Yes | Yes | Yes | Yes | Yes | Yes | Yes | Yes | | | | 8 | 100 | High |
| Pilmis, *et al* | CS | Yes | Yes | Yes | Yes | No | No | Yes | Yes | | | | 6 | 75 | Medium |
| Kizilates, *et al* | CS | Yes | Yes | Yes | Yes | No | No | Yes | Yes | | | | 6 | 75 | Medium |
| Markovska, *et al* | CC | Yes | Yes | No | Yes | Yes | Yes | No | No | Yes | UC | | 6 | 60 | Medium |
| Kibwana, *et al* | CS | Yes | Yes | Yes | Yes | Yes | Yes | Yes | Yes | | | | 8 | 100 | High |
| Godonou, *et al* | CS | Yes | Yes | Yes | Yes | Yes | Yes | Yes | Yes | | | | 8 | 100 | High |
| Kiddee, *et al* | C | Yes | Yes | Yes | Yes | Yes | No | Yes | Yes | No | Yes | Yes | 9 | 82 | High |
| Babu, *et al* | CC | Yes | Yes | Yes | Yes | Yes | No | No | Yes | Yes | Yes | | 8 | 80 | Medium |
| Hamprecht, *et al* | CS | Yes | Yes | Yes | Yes | Yes | Yes | Yes | Yes | | | | 8 | 100 | High |
| Mittal, *et al* | CC | Yes | Yes | UC | Yes | Yes | Yes | Yes | Yes | Yes | Yes | | 9 | 90 | High |
| Yan, *et al* | C | Yes | Yes | Yes | No | No | No | Yes | Yes | No | Yes | Yes | 7 | 64 | Medium |
| McConville, *et al* | C | Yes | Yes | Yes | Yes | Yes | No | Yes | Yes | No | Yes | Yes | 9 | 82 | High |
| Tran, *et al* | CS | Yes | Yes | Yes | Yes | Yes | Yes | Yes | Yes | | | | 8 | 100 | High |
| Al Fadhli, *et al* | CS | Yes | Yes | Yes | Yes | No | No | Ye | Yes | | | | 6 | 75 | Medium |

CC = Case-control, CS = Cross-sectional, Cohort, UC = Unclear.

included studies. Finally, 20 studies were found to be eligible for full-text evaluation, and that fulfilled the eligibility criteria were included in the final meta- analysis. From the included studies, five of them were case-control studies, four of them were prospective cohort and the others were cross-sectional studies. Regarding their sample size, the smallest sample size was 100 by Abdallah, H.M, *e t al* in 2017 and the largest sample size was 4376 a study by Hamprecht, *et al* in 2016. From the total 20 studies, 15 studies investigated ESBL-PE carriage rate, 5 CRE carriage rate, 5 both ESBL-PE and CRE studies. All populations included in this study were hospitalized patients with different clinical conditions and who stayed more than or equal to 4 days.

Most of the studies used double disk synergy test, Combination disk test and PCR for the confirmation of ESBL and CRE production, however, studies from Ethiopia were not used PCR. In addition, multidrug-resistance (MDR) was assessed. MDR indicates the resistance of bacteria for at least one antibiotic in different class of antibiotics, but six studies did not clearly indicate MDR in their studies. In addition, all studies use rectal swab or stool specimen. Most of the publication year of the studies were in the year of 2016 (30%), 2019 (20%) and 2021 (20%) respectively as indicated in (Table 2). Large proportion of studies were included from Asia (44%) followed by Africa (33%) as shown in (Fig 2).

## 3.2. Laboratory methods used to confirm ESBL and CRE production by *Enterobacteriaceae*

In this meta-analysis, we observed that most studies used combination disk test (CDT) and double disk synergy test (DDST) while a few studies include E-test and PCR to screen ESBL-PE and CRE producing *Enterobacteriaceae*.

**Table 2. Characteristics of individual studies included in the current systematic and met-analysis.**

| No | Author | Year | Size | Age | Design | Confirmation | Country | ESBLcarriage | ESBL-*E. coli* | ESBL-K.pn | MDR |
|---|---|---|---|---|---|---|---|---|---|---|---|
| 1 | Desta, *et al* [24] | 2016 | 267 | All | CS | CDT | Ethiopia | 139 (52%) | 106(39.7%) | 44(16.5%) | 99% (150/151) |
| 2 | Baljin,*et al* [25] | 2021 | 158 | All | C | CDT+PCR | Mongolia | 110 (69.6%) | 87 (55%) | 15 (9.5%) | 66% (85/128) |
| 3 | Aklilu, *et al* [26] | 2019 | 421 | All | CS | CDT | Ethiopia | 146(34.7%) | 62 (14.7%) | 60 (14.3) | 71% (99/140) |
| 4 | Mahamat,*et al* [27] | 2019 | 100 | All | CC | DDST+PCR | Chad | 51 (51%) | 35 (35%) | 11 (11%) | 90% (46/51) |
| 5 | Abdallah [28] | 2017 | 100 | All | CS | VITEK+PCR | Egypt | 68 (68%) | 51 (51%) | 19 (19%) | 30%(22/73) |
| 6 | Ouédraogo,*et al* [29] | 2017 | 113 | All | CC | DDST+PCR | Burkina | 47 (42%) | 34 (30.1%) | 7 (6.2%) | 68% (47/69) |
| 7 | Ebrahimi, *et al* [30] | 2016 | 4343 | All | CS | DDST+PCR | Hungaria | 323 (7.4%) | 179 (4.1%) | 185 (4.3) | |
| 8 | Pilmis, *et al* [31] | 2018 | 554 | ≥60 | CS | DDST+CDT | French | 98 (17.7%) | 69 (12.6%) | 14 (2.5%) | |
| 9 | Kizilates, et al [32] | 2021 | 168 | All | CS | Vitek+E+PCR | Turkey | 67 (39.8%) | 39 (23.2%) | 28(16.6%) | |
| 10 | Markovska, *et al* [33] | 2021 | 311 | All | CC | DDST+PCR | Bulgaria | 98 (30.2%) | 68 (21.8%) | 20 (6.4%) | |
| 11 | Kibwana,*et al* [34] | 2020 | 196 | ≥18 | CS | CDT | Tanzania | 117 (59.7) | 80 (40.8%) | 37(18.9%) | 89%(117/131) |
| 12 | Godonou, *et al* [35] | 2022 | 105 | All | CS | CDT | Togo | 85 (80.9%) | 73 (69.5%) | 24(22.9%) | 84% (85/101) |
| 13 | Kiddee, *et al* [36] | 2019 | 215 | All | C | CDT+PCR | Thailand | 134 (62.3%) | 108(50.2%) | 31(14.4%) | 98%(156/160) |
| 14 | Babu, *et al* [37] | 2016 | 220 | ≥18 | CC | CDT+PCR | India | 138 (62.7%) | 97(44.1%) | 37(16.8%) | 50%(61/134) |
| 15 | Hamprecht,*et al* [38] | 2016 | 4376 | ≥18 | CS | CDT+PCR | German | 416 (9.5%) | 344 (7.9%) | 37 (0.9%) | |
| | **Colonization rate of CRE** | | | | | | | **Overall CRE** | **CRE-*E. coli*** | **CRE-K. pn** | |
| 1 | Desta, *et al* [24] | 2016 | 267 | All | CS | CDT | Ethiopia | 5 (2%) | 3 (1.1%) | 2 (0.8%) | |
| 2 | Mittal, *et al* [39] | 2016 | 200 | All | CC | DDST+PCR | India | 71 (35.5%) | 30 (15%) | 9 (4.5%) | 54 (27/50) |
| 3 | Yan, *et al* [40] | 2020 | 150 | All | C | Vitek +PCR | China | 25 (16.6%) | 6 (4.0%) | 17(11.3%) | 96% (25/26) |
| 4 | McConville, *et al* [41] | 2017 | 338 | ≥ 18 | C | Vitek | Colombia | 94 (28%) | 39 (11.5%) | 42(12.4%) | 97% (35/36) |
| 5 | Tran, *et al* [42] | 2019 | 2233 | All | CS | Vitek | Vietnam | 1165 (52%) | 682(30.5%) | 805(36.%) | |
| 6 | AlFadhli, *et al* [43] | 2020 | 590 | ≥18 | CS | PCR | Kuwait | 58 (9.8) | 11 (1.9%) | 29 (4.9%) | 100% (56/56) |
| 7 | Abdallah [28] | 2017 | 100 | All | CS | CDT+PCR | Egypt | 5 (5%) | | 5 (5%) | |
| 8 | Kizilates, *et al* [32] | 2021 | 168 | All | CS | Vitek+E+PCR | Turkey | 21 (12.5%) | 10(47.6%) | 11(52.4%) | |
| 9 | Markovska, *etal* [33] | 2021 | 311 | All | CC | DDST+PCR | Bulgaria | 3 (1%) | | 3 (1%) | |
| 10 | Hamprecht,*et al* [38] | 2016 | 4376 | ≥18 | CS | CDT+PCR | German | 5 (0.11) | | | |

CC = Case-control, CS = Cross-sectional, C = Cohort, CDT = Combination disk test, DDST = double disk synergy test, E = E-test.

### 3.3. The pooled estimate of ESBL-PE and CRE Colonization among hospitalized patients

Separate analysis was made for ESBL-PE and CRE colonization. Accordingly, the overall pooled estimate of ESBL-PE was 45.6% (95% CI: 34.11–57.10, P< 0.001) with high level of heterogeneity ($I^2$ = 99.63%) as indicated in (Fig 3). The pooled estimate of CRE colonization was 16.19% (95% CI: 5.46–26.91, P <0.001) with high level of heterogeneity ($I^2$ = 99.78%) as demonstrate in (Fig 4). The predominant ESBL producer was *E. coli*, 32.99% (95% CI: 23.28–42.69, P<0.001) followed by *K. pneumoniae*, 11% (95% CI: 7.98–14.89, P<0.001) as shown in (Figs 5 and 6) respectively. In addition, the most common CRE producer was *K. pneumoniae*, 50% (95% CI: 25.09–73.96, P<0.001) in (Fig 7), and the least common CRE producer was *E. coli*, *43%* (95% CI: 27.93–58.68, P<0.001) as depicted in (Fig 8).

### 3.4. Meta-analysis of publication bias

Publication bias was assessed to investigate problems related to published and unpublished papers, the analysis showed assymetric funnel plot which indicates the presence of publication bias. Similarly, the Egger's test showed a significant publication bias (p<0.001) as indicated in (Fig 9) and (Table 3).

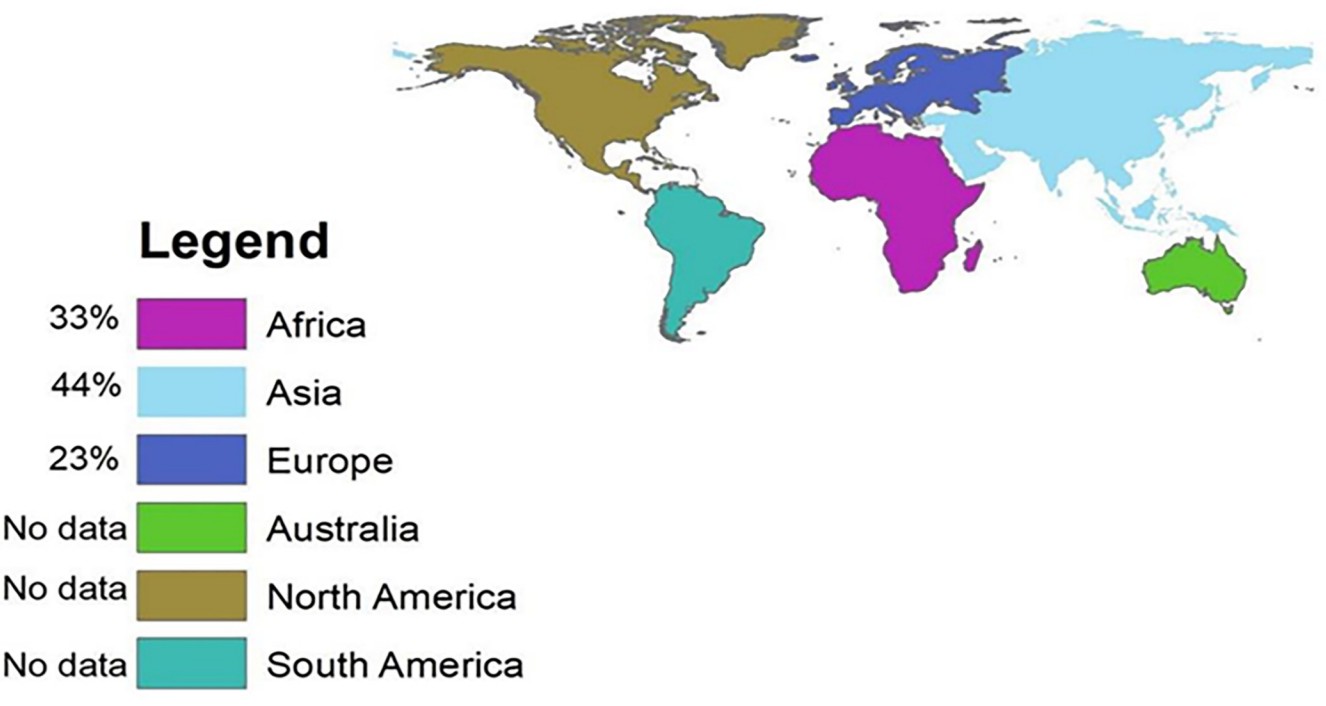

**Fig 2. Global regions included in this study.**

### 3.5. Subgroup analysis of ESBL-PE and CRE colonization by region

Sub group analysis of ESBL-PE and CRE colonization was performed by geographical locations. The highest pooled estimate of ESBL-PE was reported in Asian region,58.23% (95% CI: 45.79–70.67, P<0.001, $I^2$ = 91.96%) followed by the African region,55.36% (95% CI: 43.68–67.03, P<0.001, $I^2$ = 94.42%,). The least pooled estimate was recorded from Europe region, 15.96% (95% CI: 6.01–25.91, P<0.001, $I^2$ = 99.52%,) as indicated in (Fig 10). When stratified by country, the top four ESBL-PE prevalence were reported in Togo 80.90% (95% CI: 72.98–88.82), Mongolia 69.60% (95% CI: 62.19–77.01), Egypt 68.0% (95% CI: 58.52–77.48), Thailand 62.30% (95% CI: 55.65–68.95) and India 61.0% (95% CI: 54.38–67.61) (Fig 10).

Similarly, CRE colonization subgroup analysis was made and relatively higher pooled prevalence was found in Asia than Europe and Africa, 28.43% (95% CI: 13.78–43.07, P<0.001, $I^2$ = 98.77), 4.12% (95% CI: -3.27–11.51, P<0.001, $I^2$ = 98.90) and 2.64% (95% CI: 0.23–5.04, P = 0.28, $I^2$ = 15.76) respectively as depicted in (Fig 11). High CRE colonization was reported in Vietnam, 52.0% (95% CI: 49.90–54.10).

### 3.6. Subgroup analysis of ESBL-PE and CRE by study design

We also performed subgroup analysis by study type as indicated in (Fig 12). The highest pooled estimated of ESBL-PE was found in the cohort study, 65.78% (95% CI: 58.62–72.90, P = 0.001, $I^2$ = 51.58%). Whereas case-control and cross-sectional studies showed low pooled estimates with the highest heterogeneity, 45.91% (95% CI: 32.0–59.23, P = 0.001, $I^2$ = 92.16%) and 36.14% (95% CI: 19.07–53.21, P = 0.001, $I^2$ = 99.85%) respectively.

When we stratified CRE colonization by study design, the pooled estimate was reported at 22.52% in cohort study (95% CI: 11.45–33.59, P = 0.001, $I^2$ = 87.13%), 18.08% in case-control (95% CI: -15.73–51.89, P = 0.001, $I^2$ = 98.94%) and 13.56% in cross-sectional study (95% CI: -2.04–29.16, P = 0.001, $I^2$ = 99.76%) as shown in (Fig 13).

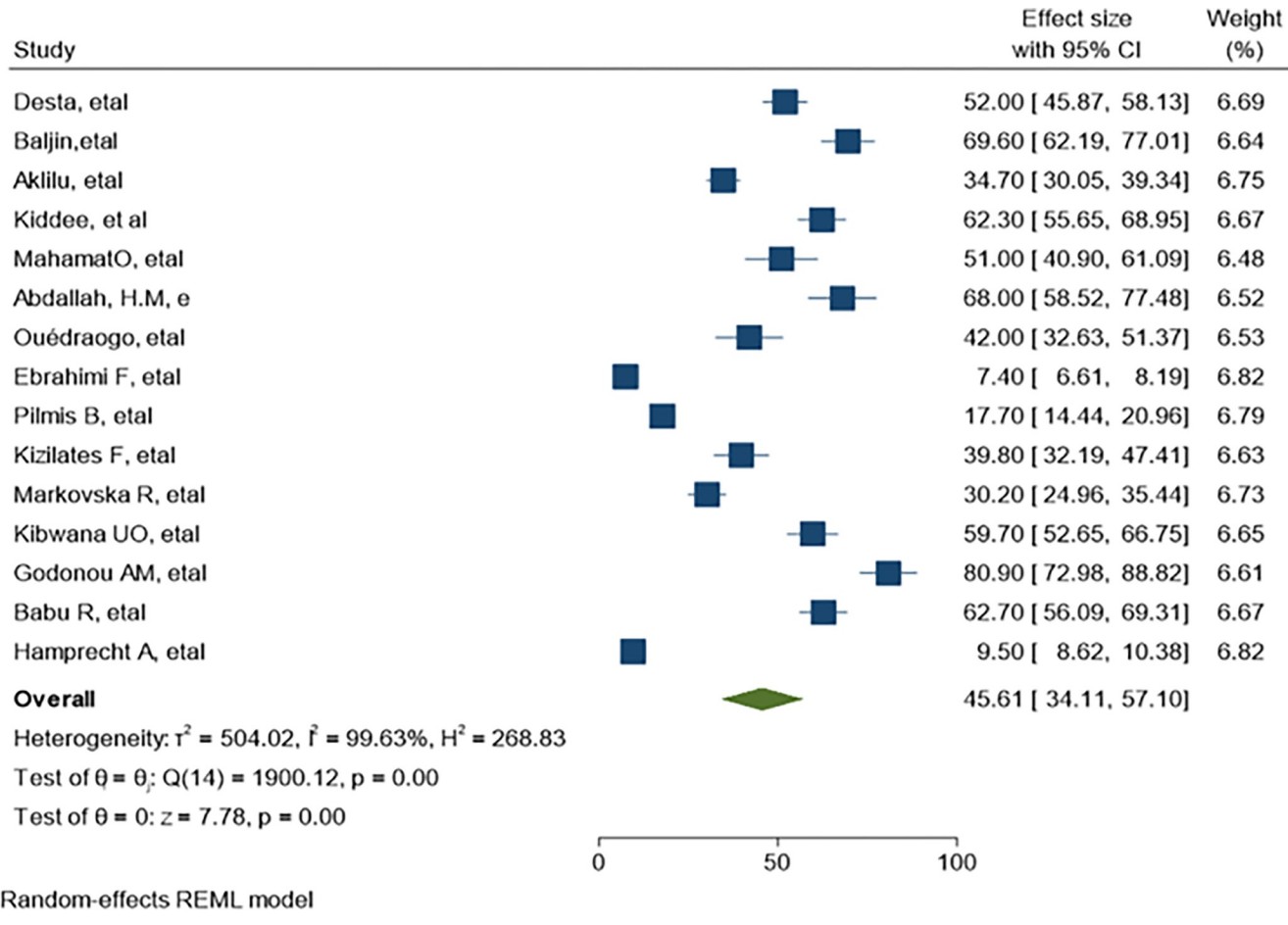

**Fig 3. Forest plot for ESBL-PE colonization of hospitalized patients.**

### 3.7. Pooled estimate of MDR

Only 13 studies were eligible for multi drug resistance (MDR) meta-analysis and the pooled estimate was 76% (95% CI: 64.29–88.48, P<0.001) as shown in (Fig 14). Most of the studies defined MDR as bacteria resistant to two or three class of antimicrobials. The highest and lowest MDR was recorded at 99.3% (95% CI: 97-22-101.38) and 45.5% (95% CI: 36.82–54.18) respectively.

### 3.8. Pooled estimate of odds ratio for history of hospitalization

The majority of studies defined previous hospitalizations as admission within the last 3, 6 or 12 months. Five studies analyzed to figure out whether previous hospitalization can be a risk factor or not for fecal colonization with ESBL-producing *Enterobacteriaceae*, and we found a 0.93 pooled estimate odds ratio (95% CI: 0.54–1.32) as indicated in (Fig 15). Five studies were eligible for meta-analysis to assess the impact of prolonged hospitalization for ESBL colonization. These studies defined hospital stay as patients admitted for ≥ 7 days, and the analysis reported that the odds of 0.94 (95% CI: 0.52–1.37) as indicated in (Fig 16).

### 3.9. Pooled estimate of odds ratio for history of antibiotic use

History of antibiotic treatment was analyzed in seven studies, and a history of antibiotic treatment in the last 3 and 6 months was found to be a risk factor for ESBL-PE colonization. The

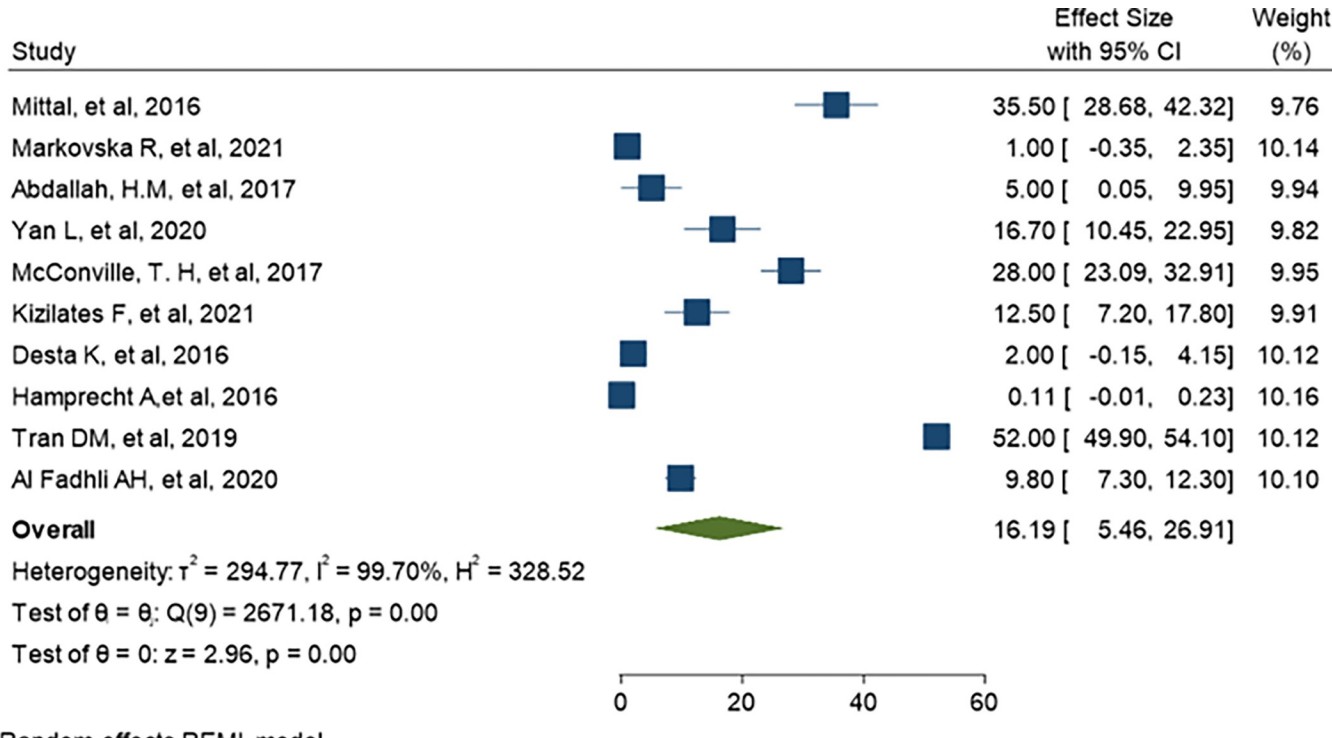

**Fig 4. Forest plot for CRE colonization rate of hospitalized patients.**

pooled estimate was 1.45 (95% CI: 0.81–2.08) as shown in (Fig 17). In addition, current antibiotic use during admission such as carbapenem and fluoroquinolone was assessed among two studies. Carbapenem treatment found to be associated with the colonization of CRE (OR:1.72, 95% CI:1.05–2.40) while fluoroquinolone (OR:0.98, 95% CI:0.59–1.38) was not associated as indicated in (Figs 18 and 19).

### 3.10. Meta-analysis of the association between comorbidities and ESBL-PE or CRE colonization

As indicated in (Table 4), meta-analysis was performed to assess the association of comorbidities with ESBL-PE or CRE colonization. Accordingly, two studies assessed for any chronic diseases and showed an increased risk for ESBL colonization, 2.02 (95% CI: 1.17–2.87). Related to renal diseases, three studies were analyzed and associated with CRE colonization with the odds of 1.78 (95% CI: 0.06–3.50), and another two studies with diabetics showed that CRE colonization can be a risk factor with 1.42 (95% CI: 0.20–2.65). In addition, two cohort studies were separately analyzed to assess the CRE colonization rate at the time of admission and after ≥7 days of admission. The result showed that there was an increased rate of CRE colonization at admission and after admission with odds of 14.77 (95% CI: -1.35–30.90) and 45.63 (95% CI: -0.86–92.12) respectively.

## 4. Discussion

Our systematic review and meta-analysis assessed factors influencing colonization with MDRB in different studies across the globe. We analyzed pooled estimate of MDR, ESBL-PE and CRE colonization, comorbidities, and risk factors of MDR colonization reported from

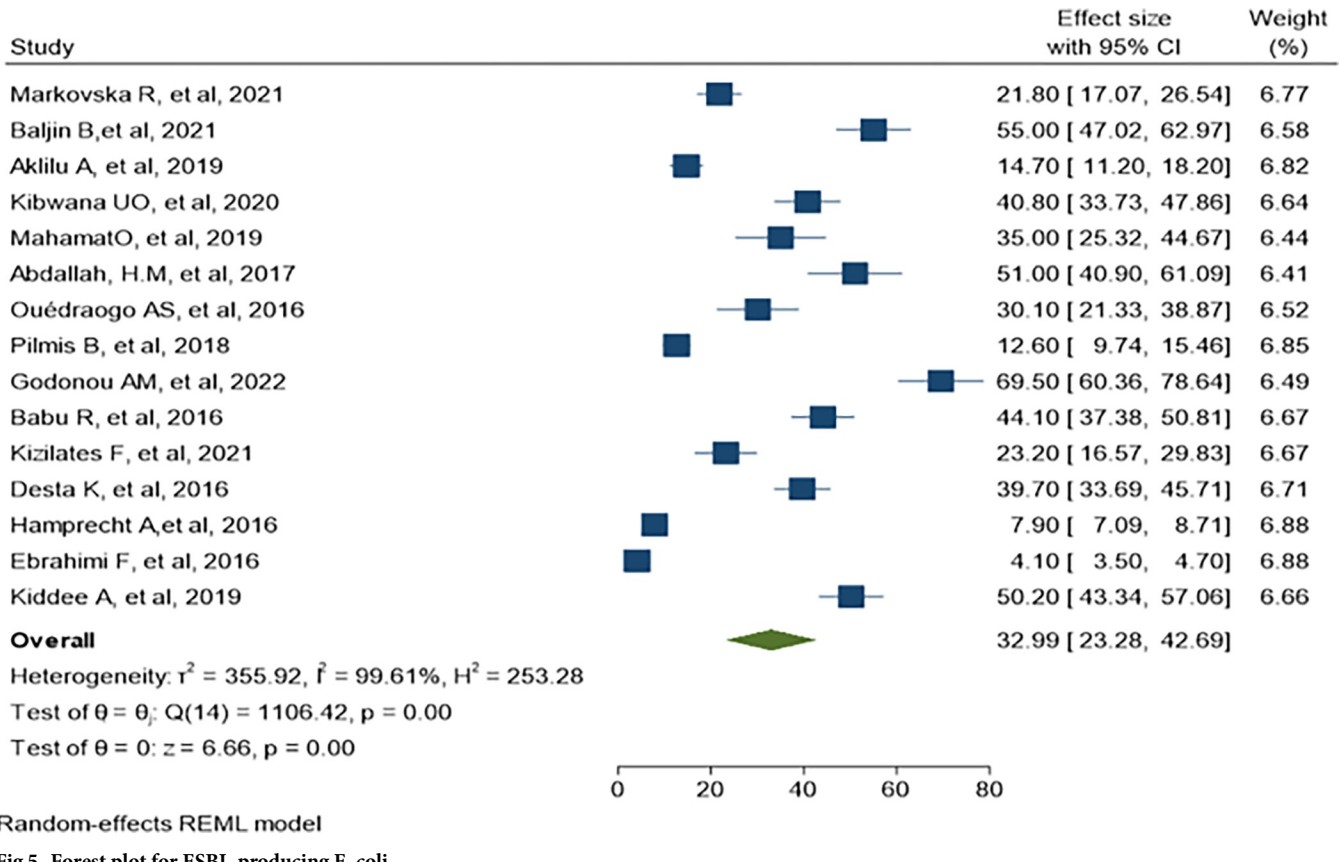

**Fig 5. Forest plot for ESBL-producing E. coli.**

January1,2016-August 10, 2022. It also focused on *E. coli* and *K. pneumoniae* because these bacteria are identified as MDR bacteria that are strongly associated with nosocomial infection [44]. In this study, the source of specimen was stool /rectal swab among hospitalized patients (ICU or general ward patients).

The emergence of multidrug-resistant bacteria has become a global priority concern [45]. Particularly, ESBL-PE and CRE are highly linked to prolonged hospitalization, morbidity, mortality and extra treatment cost [46–52]. In the present study, a meta-analysis compiled from different countries revealed a pooled estimate of ESBL-PE colonization among hospitalized patients, 45.6% (95% CI: 34.11–57.10) with the lowest prevalence in Hungary,7.4% and the highest reported in Togo, 80.9% [30,35] respectively. The highest report in Togo could be due to its' recent publication period, which was in 2022. Our finding was higher than from a systematic study of sub-Saharan Africa on admitted patients with a pooled prevalence of 32% [53]. This variation might be due to geographical location, our study focused on three continents (Africa, Asia and Europe) which could increase the prevalence of ESBL-PE colonization compared to the sub-Saharan Africa study. Similarly, lower result was also reported from another systematic review, 18% [54] that was mainly from European countries conducted before 2016. This gap could be due to study period and geographical location.

In this meta-analysis, the predominant ESBL-PE were *E. coli* and *K. pneumoniae* with pooled estimate of 32.99% and 11% respectively. These figures were much lower than studies in Togo (69.5% vs22.9%), India (56% vs 43%), and Ethiopia (54.9% vs 33.5%) [35,55,56] respectively. Similarly, a large prospective study showed high prevalence of *E. coli*,57% and *K.*

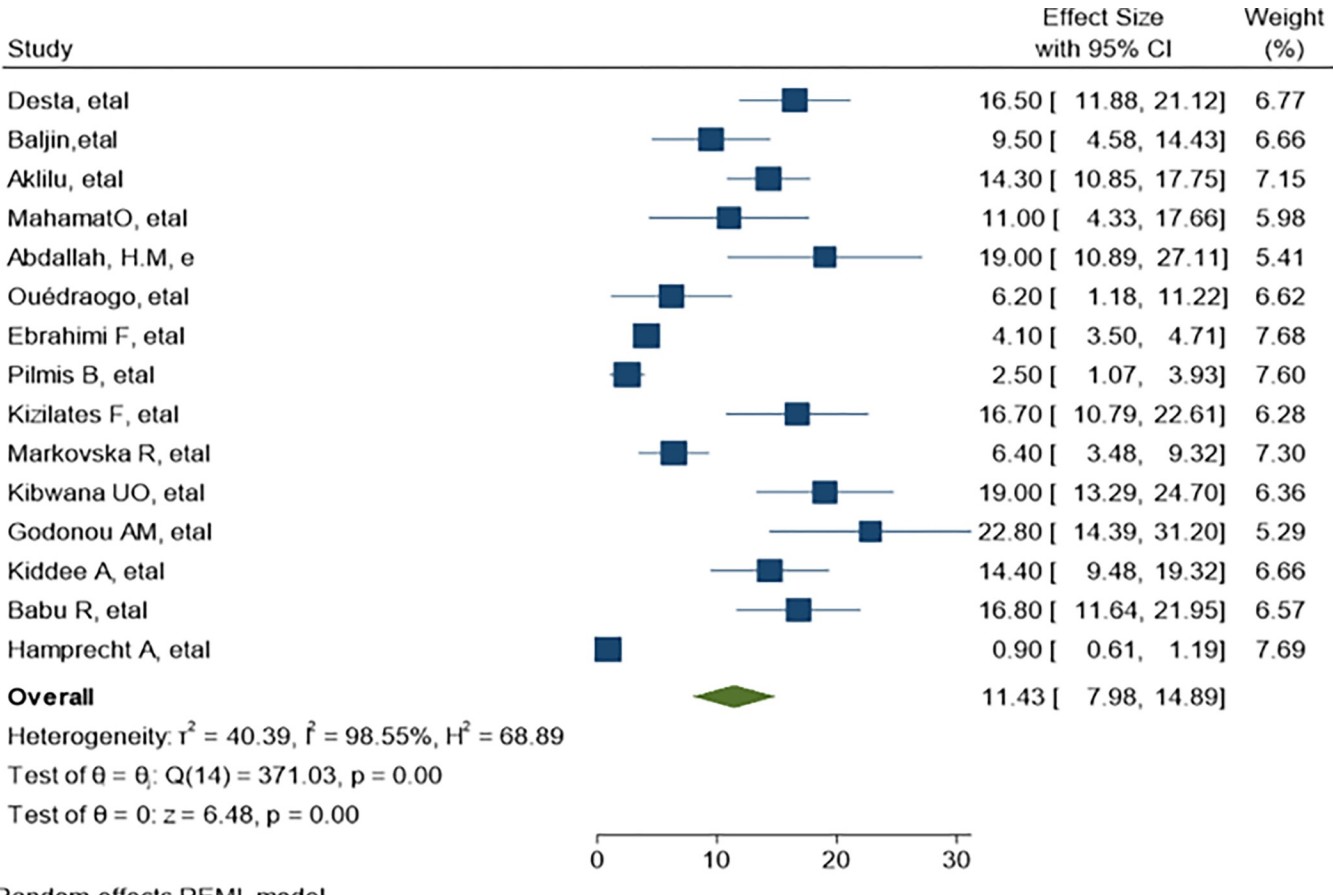

**Fig 6. ESBL-producing K. pneumoniae.**

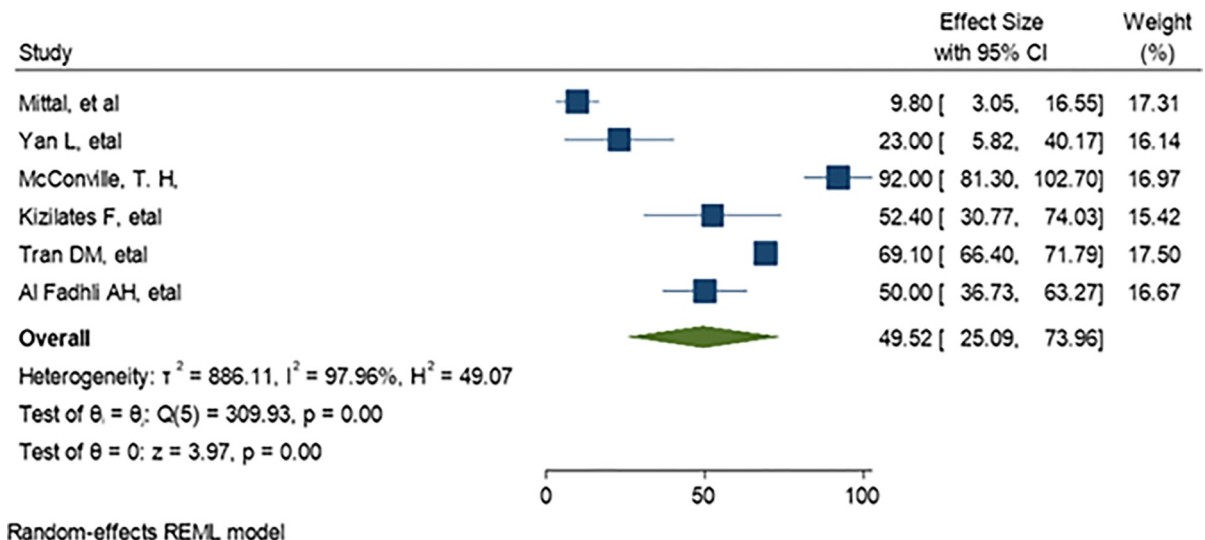

**Fig 7. Forest plot showing the estimate of carbapenem producing K. pneumoniae.**

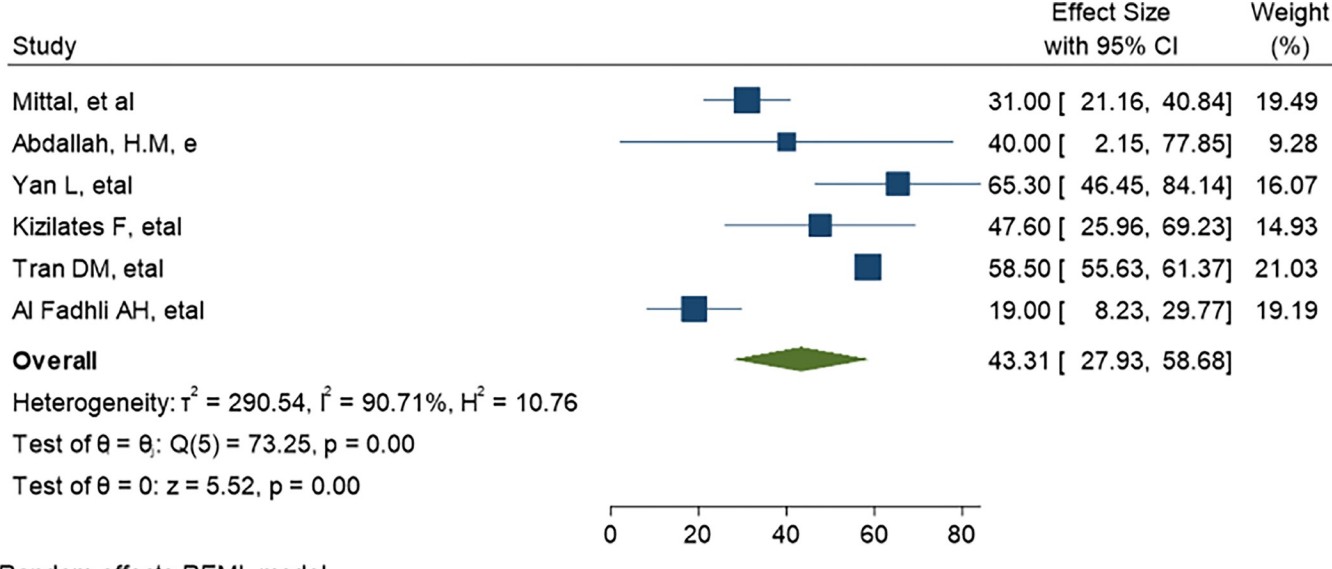

**Fig 8. Forest plot showing the estimate of Carbapenem producing E.coli.**

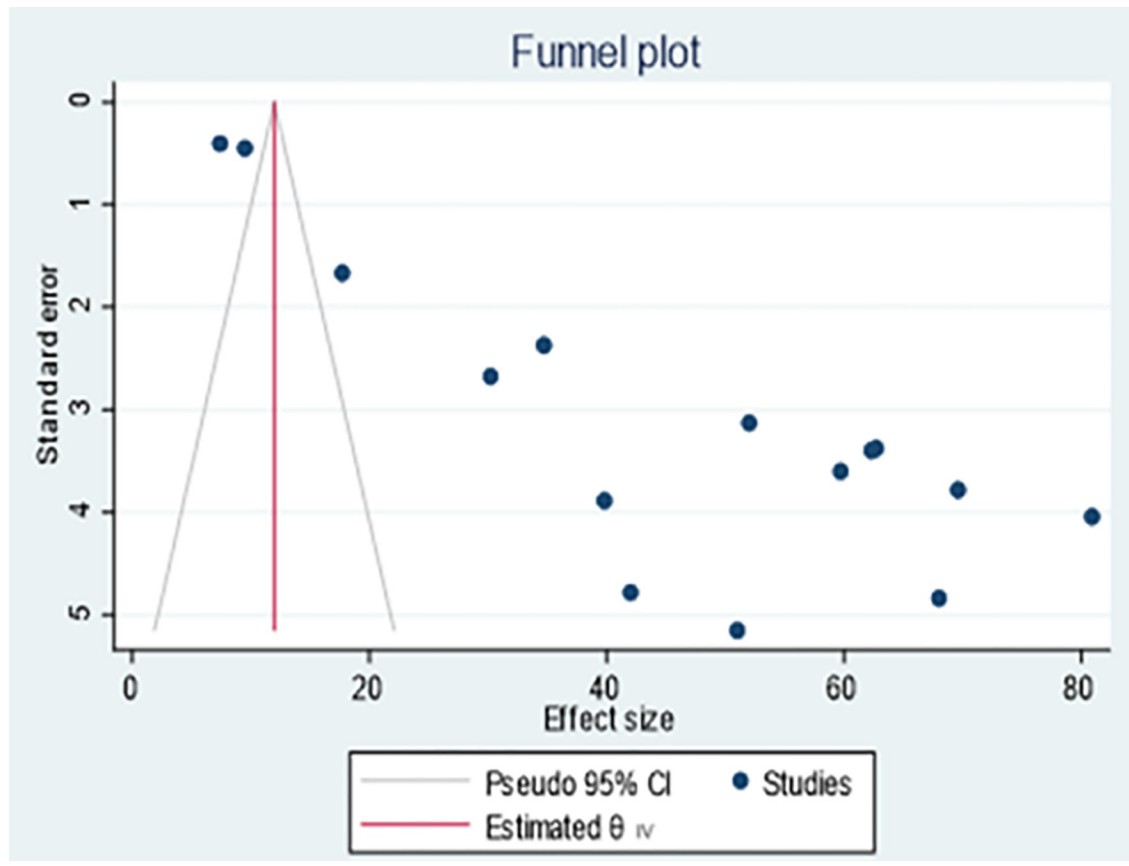

**Fig 9. Funnel plot showing publication biases.**

**Table 3. Publication bias and egger's test of ESBL-PE and CRE pooled estimates.**

| Category | Number of studies | Pooled estimate | | | Eggers regression test, p-value |
|---|---|---|---|---|---|
| | | Estimate (prevalence), 95%CI | Heterogeneity | | |
| | | | $I^2$ | P-value | |
| ESBL-PE colonization | 15 | 45.6% (34.11–57.10) | 99.6 | <0.001 | <0.001 |
| CRE colonization | 10 | 16.19% (5.46–26.91) | 99.7 | <0.001 | <0.001 |

*pneumoniae*,43% [57]. However, *K. pneumoniae* was reported as a dominant ESBL-PE with a prevalence of 51.7% over *E. coli*,46.4% [58]. This inconsistent finding might be related with study setting variations and detection methods. In the current meta-analysis, the pooled estimate of CRE colonization was 16.19% (95% CI: 5.46–26.91), which was much greater than a previous systematic review and meta-analysis study that reported CRE colonization at 3.0% mainly from America and Europe [59]. This greater prevalence in our meta-analysis may be due to geographical locations, most of included studies were from African and Asian regions. Another possible reason might be due to number of studies included, we included 10 studies with CRE colonization while they included 4 studies. In contrast, our finding was lower than studies in Thailand, 69.7% [60], India, 18% [61] and Egypt,28% [62]. This disparity could be due to our result was an average from different countries, and another reason may be related with the presence of serious comorbidities, medical intervention and overuse of antibiotics in their study. From the total pooled carbapenem resistant *Enterobacteriaceae*, *K. pneumoniae* accounted 50% and *E. coli* 43%. This was inconsistent with study from India, *E. coli* 71.4%, *K. pneumoniae* 23.8% [63], and another report from Mexico showed *E. coli* 72.7% and *K. pneumoniae* 23.6% [64].

In this study, subgroup meta-analysis was performed which showed the highest pooled prevalence of ESBL-PE colonization in Asia, 58.23% (95% CI: 45.79–70.67). This high prevalence might be related to the high consumption of beta-lactam antibiotics, leading to the emergence of plasmid-mediated resistance to beta-lactam antibiotics [65]. Another suggested reason was low control mechanism in antibiotic usage which causes overuse and misuse of antibiotics in the healthcare facilities and animal husbandry, and considered as highly vulnerable region for antimicrobial resistance [66,67]. Africa was the second most common region that reported significant ESBL-PE colonization rate,55.36% (95% CI: 43.68–67.03). The reason for this considerable report might be due to poor antibiotic stewardship and limited resources to prevent and control the spread of multidrug-resistance [68]. The least ESBL colonization was recorded in Europe regions 15.9%, which was similar to a systematic review and meta-analysis with a prevalence of 15.0% [69]. This low prevalence could be explained by the fact that these regions are high income countries that have good infrastructure and steward ship program to prevent and control the spread of multidrug-resistance [68]. It might be also related with large sample size; estimates from large population size may lower the pooled estimate.

When we compared at country level in each continent, the highest ESBL-PE colonization was reported in Mongolia, 69.60% (95% CI: 62.19–77.01), Thailand 62.30% (95% CI: 55.65–68.95) and India 61.0% (95% CI: 54.38–67.61). This high prevalence may be related with study design, for instance Mongolian study was a cohort study which showed an increased colonization rate of ESBL-PE during their 14-day hospitalization while data were taken at the time of admission in the case of Thailand and Indian study. It could be also due to study population and study settings; the Mongolian study used all age groups at trauma center whereas in Thailand and India the study population were above the age of 18 years old.

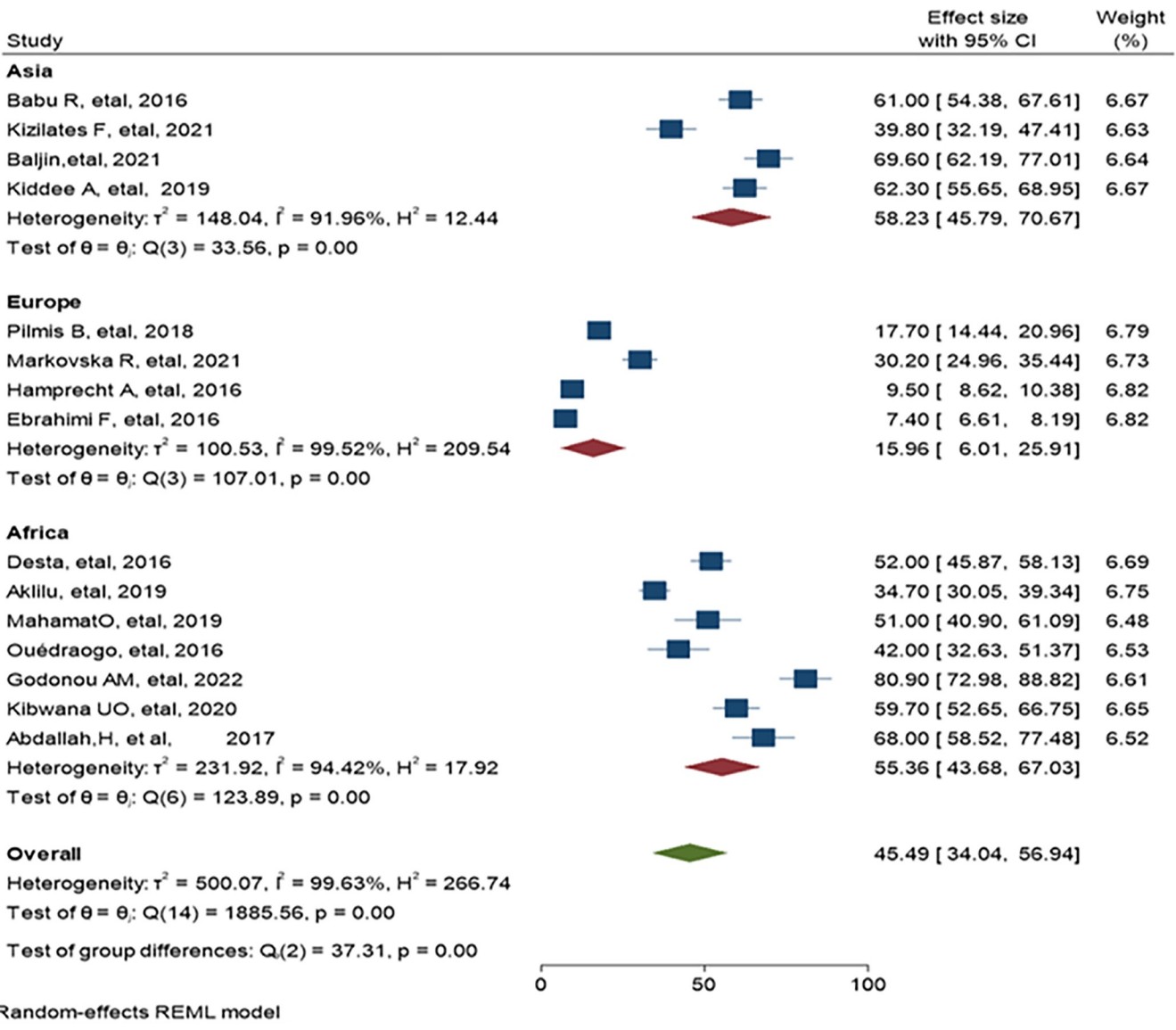

**Fig 10. Forest plot showing subgroup analysis of ESBL-PE in Asia, Europe and Africa.**

In African countries, the highest ESBL-PE colonization were reported from Togo, 80.90% (95% CI: 72.98–88.82) and Egypt, 68.0% (95% CI: 58.52–77.48). The possible reason for this difference might be due to publication period, example the data from Togo was very recent,2022 which strengthen the increasing trend of antimicrobial resistance globally [70]. Study population might be also considered as a reason for this discrepancy, for example the study in Togo used all admitted patients in all wards whereas in Egypt only gastrointestinal complaints were participated.

In the present systematic review and meta-analysis, CRE colonization subgroup analysis was performed and revealed the highest pooled estimate in Asia region, 28.43% (95% CI: 13.78–43.07). Lower colonization rate of CRE was reported in Africa and Europe regions than Asian region, this might be due to detection of CRE colonization was not their target while the Asian regions studied exclusively CRE carriage rate. When stratified by country, higher CRE

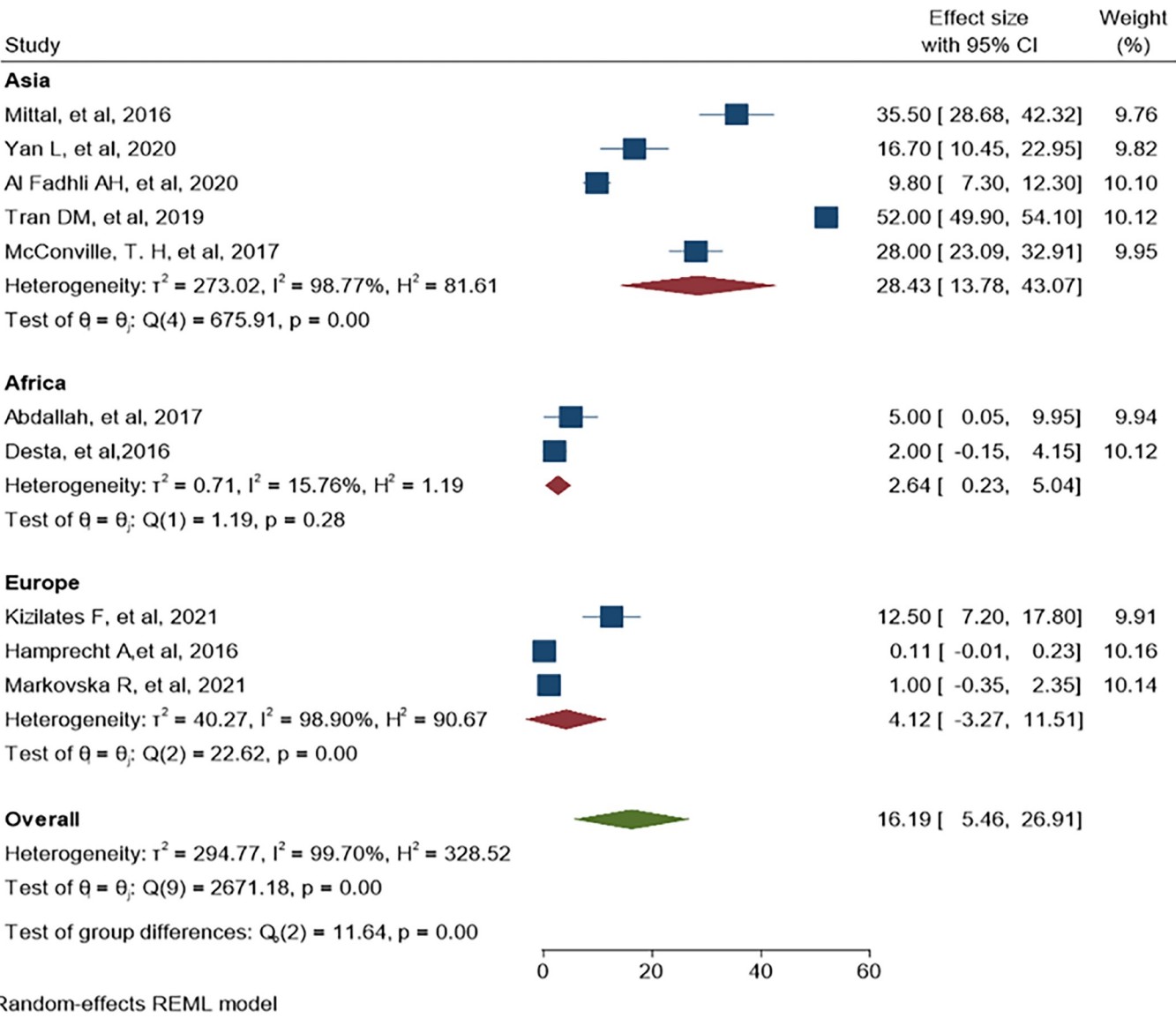

**Fig 11. Forest plot of CRE colonization in Asia, Africa and Europe region.**

colonization was reported in Vietnam, 52.0% (95% CI: 49.90–54.10) than other Asian countries. This high burden of CRE could be due to large sample size from 12 hospitals that can enhance the probability of detecting more CRE colonization. Additionally, it might be due to over use of carbapenem antibiotics in the community because of lack of control mechanisms in the usage of antibiotics. Moreover, we performed subgroup analysis by study type and found a high level of heterogeneity across study designs. The highest variability was indicated in casecontrol and cross-sectional studies, however cohort study showed medoderate heterogeneity. The highest variability among study designs might be due to recall bias and selection bias,especially in casecontrol and cross-sectional studies [71]. The largest pooled estimate of ESBL-PE and CRE colonization was recorded in cohort study which could be due to low number of studies involved in the analysis.

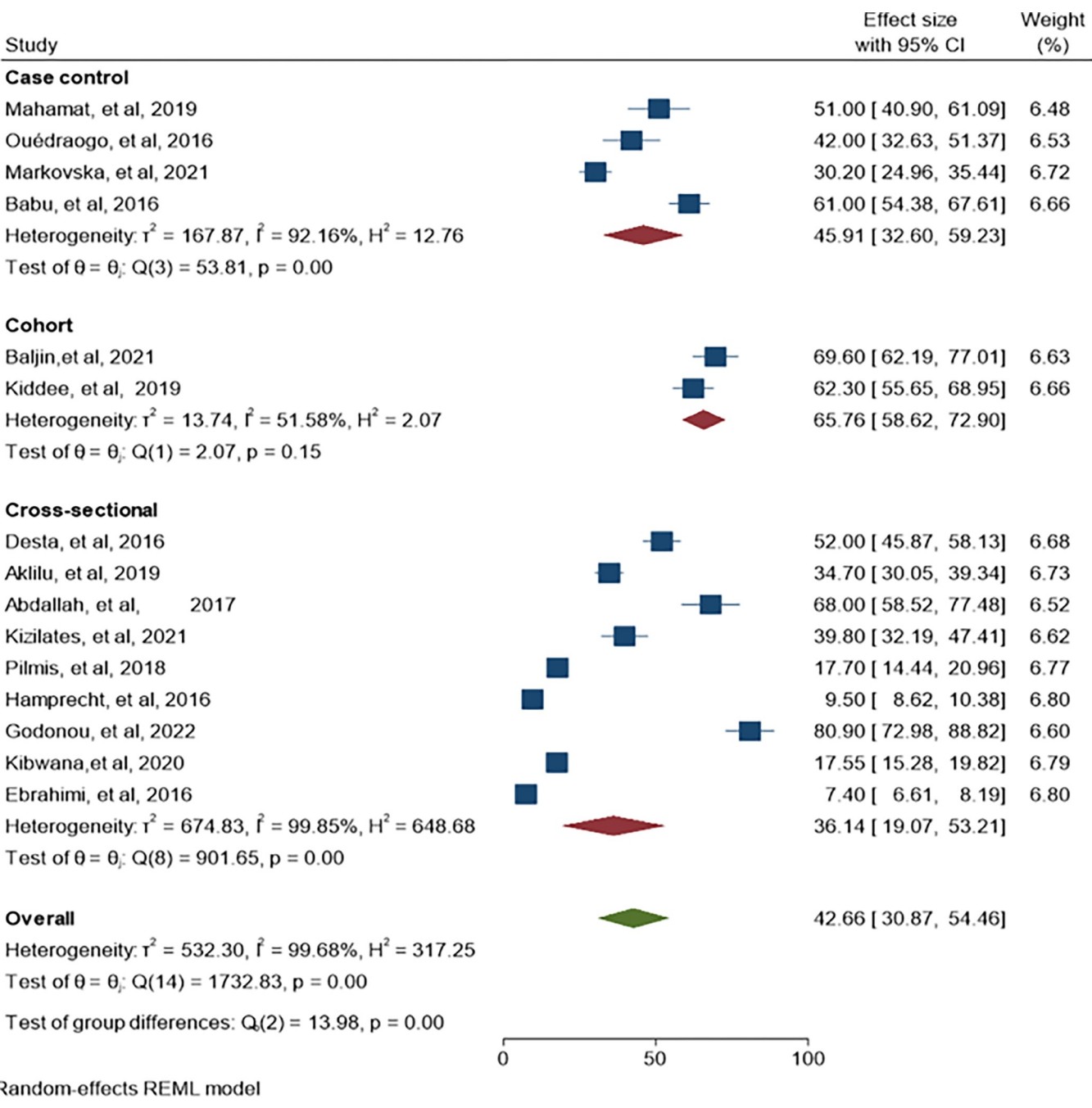

**Fig 12. ESBL-PE subgroup analysis by study design.**

Multidrug resistant bacteria are more prevalent in hospitalized patients and cause more infections because of invasive procedures, chronic diseases and over use of antibiotics [72]. The rate of multidrug-resistant bacteria range from 30–50% in china [73] and 89.5% in Africa [74]. Nearly 20% of hospital acquired infections are due to multidrug resistant bacteria [75]. In our meta-analysis, the pooled prevalence of MDR was 76% (95% CI: 64.29–88.48), which was slightly higher than a systematic and meta-analysis in Ethiopia,71% [19]. This variation may be related with geographical location, high consumption rate of antibiotics and presence of

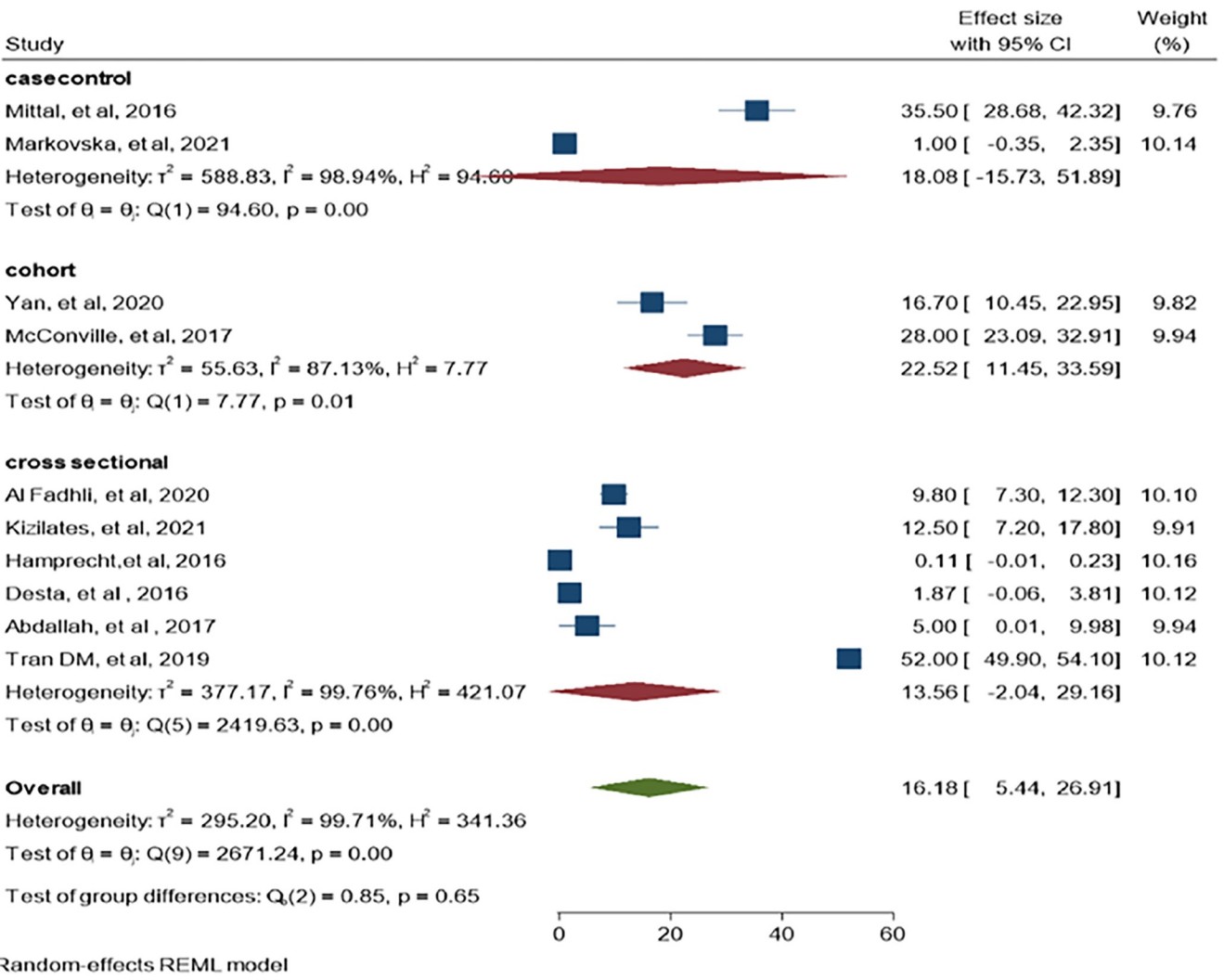

**Fig 13. CRE subgroup analysis by study design.**

sever underlining comorbidities in our study population. Another systematic review and meta-analysis from Asia reported a lower MDR prevalence 66% [76] than our finding. This difference might be due to specific study population that were with diarrheagenic patients and focused on only *E. coli* in their study. On the other hand, our finding was less than a systematic review and meta-analysis reported from Ethiopia with pooled prevalence of 82.7% [77] and a single study in Debre Brehan, Ethiopia 87.8% [56]. This variation could be due to specimen source; we used fecal specimen while they included multiple sample sources. Furthermore, misuse and overuse of antibiotics including poor implementation of antimicrobial stewardship in their study settings can contribute for the increasing rate of MDR.

Several studies identified many risk factors of multidrug-resistant bacteria colonization among hospitalized patients. Such as previous hospitalization, previous antibiotic treatment, length of hospital stays, and co-morbidities like surgery, liver cirrhosis, diabetics, and renal diseases [78–80]. In healthcare facilities, intestinal carriage of ESBL-PE is one of the main reservoirs of multidrug-resistant bacteria which have been associated with high risk for developing infections [81,82]. In our meta-analysis, six studies were eligible to assess the association of

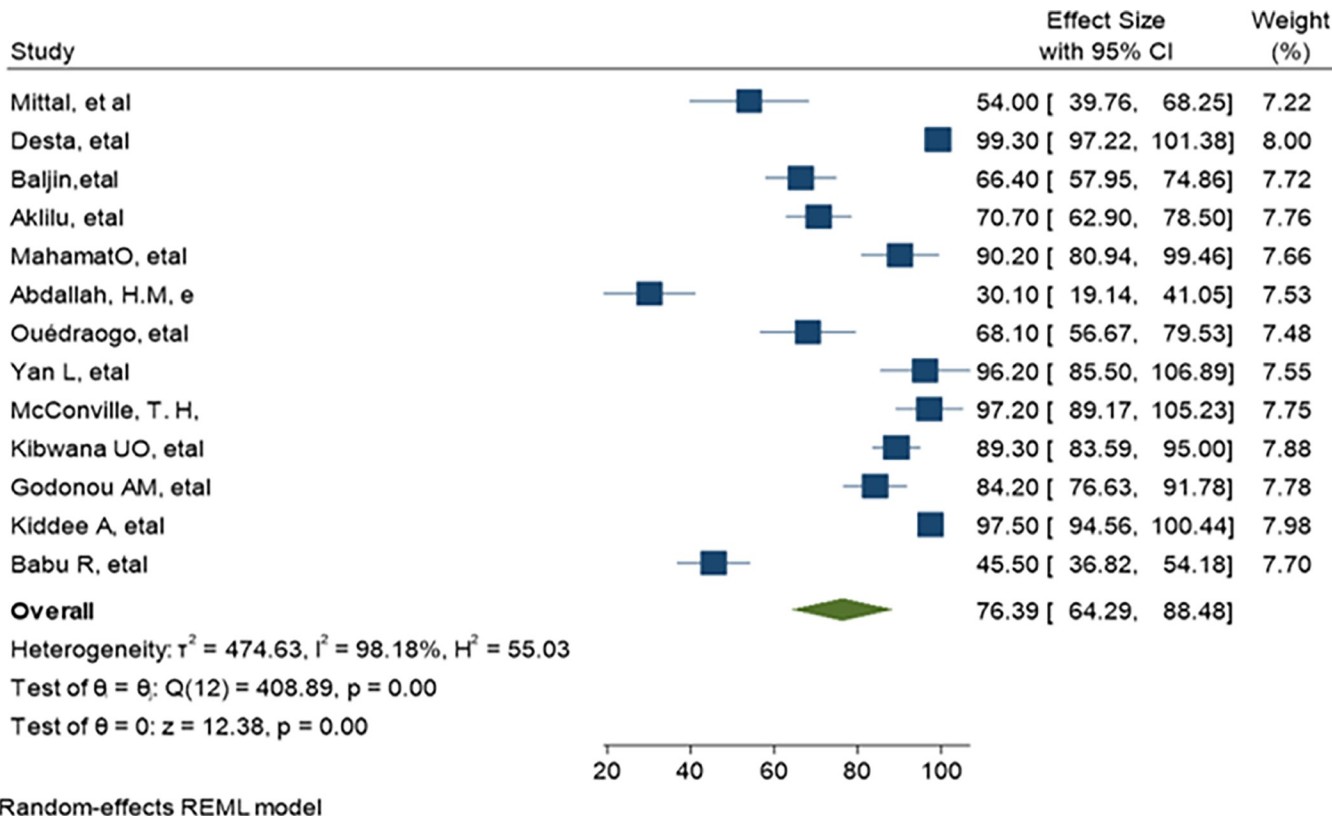

**Fig 14. Pooled estimate of Multidrug resistant level.**

previous hospitalization and antibiotic treatment with ESBL-PE colonization. The meta-analysis showed that there was no association between previous hospitalization and ESBL colonization, this lack of association might be due to the low number of studies included in the analysis. This finding disagreed with a systematic and meta-analysis study [83], and another individual study [84]. However little association was noticed between previous antibiotic treatment and ESBL colonization, which was consistent to a previous study [83]. This association might be due to the fact that antibiotic use has been strongly implicated in the development of antimicrobial resistance [85].

Meta-analysis was also performed to assess the association of comorbidities with the acquisition of ESBL-PE or CRE. Chronic diseases have showed an increased risk for ESBL-PE colonization, 2.02 (95% CI: 1.17–2.87). In addition, renal diseases and diabetics were associated with CRE colonization with the odds of 1.78 (95% CI: 0.06–3.50) and 1.42 (95% CI: 0.20–2.65) respectively. This finding was congruent with previous studies in Ethiopia [26,56], Algeria [86] and India [87]. This association may be due to the fact that immunocompromised patients are more prone to be colonized with MDRB. In the current study, colonization of CRE is increased during hospital stay with the odds of 14.77 (95% CI: -1.35–30.90) at the time of admission and 45.63 (95% CI: -0.86–92.12) after ≥7 days of hospital stay, this was consistent with previous studies [25,78].

## 5. Limitation of the study

We used only three data bases, this can limit to access full information about published articles. Risk factor estimation was based on limited number of studies, due to inconsistent reports. In addition, articles published other than English were not included.

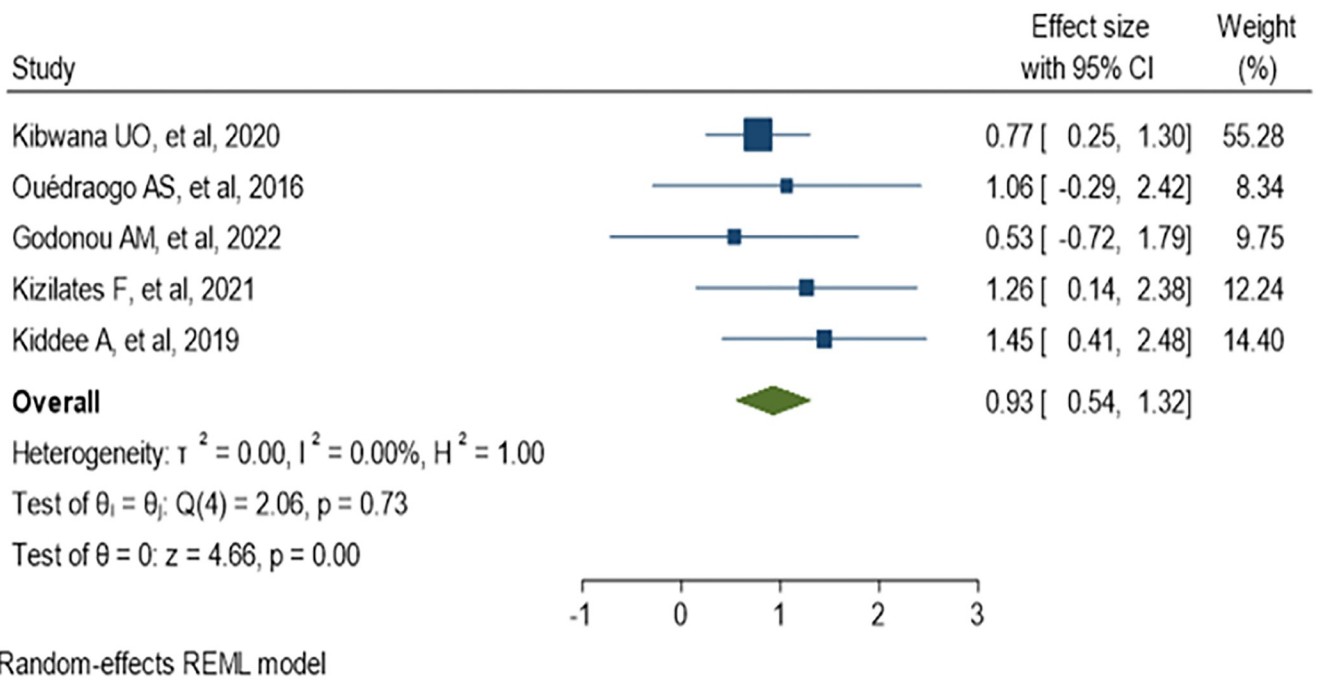

**Fig 15. Forest plot of meta-analysis of ESBL-PE colonization with previous hospitalization.**

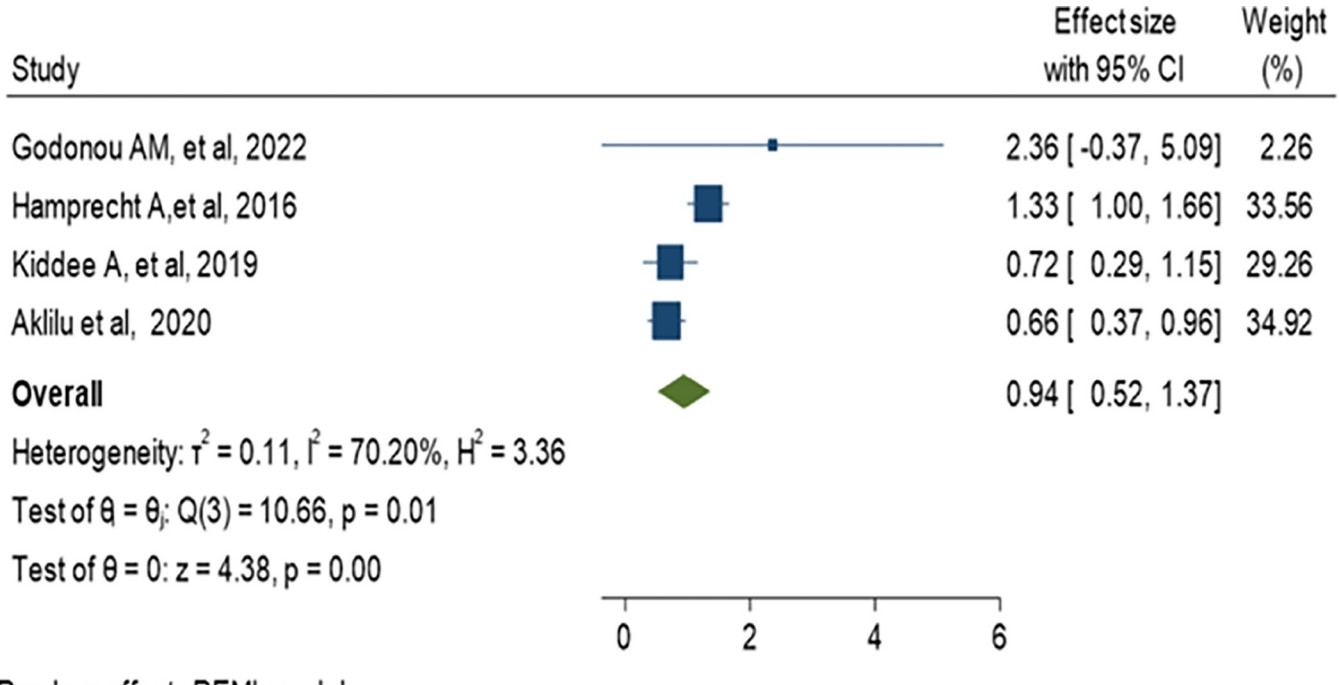

**Fig 16. Forest plot for meta-analysis of hospital stay with ESBL-PE colonization.**

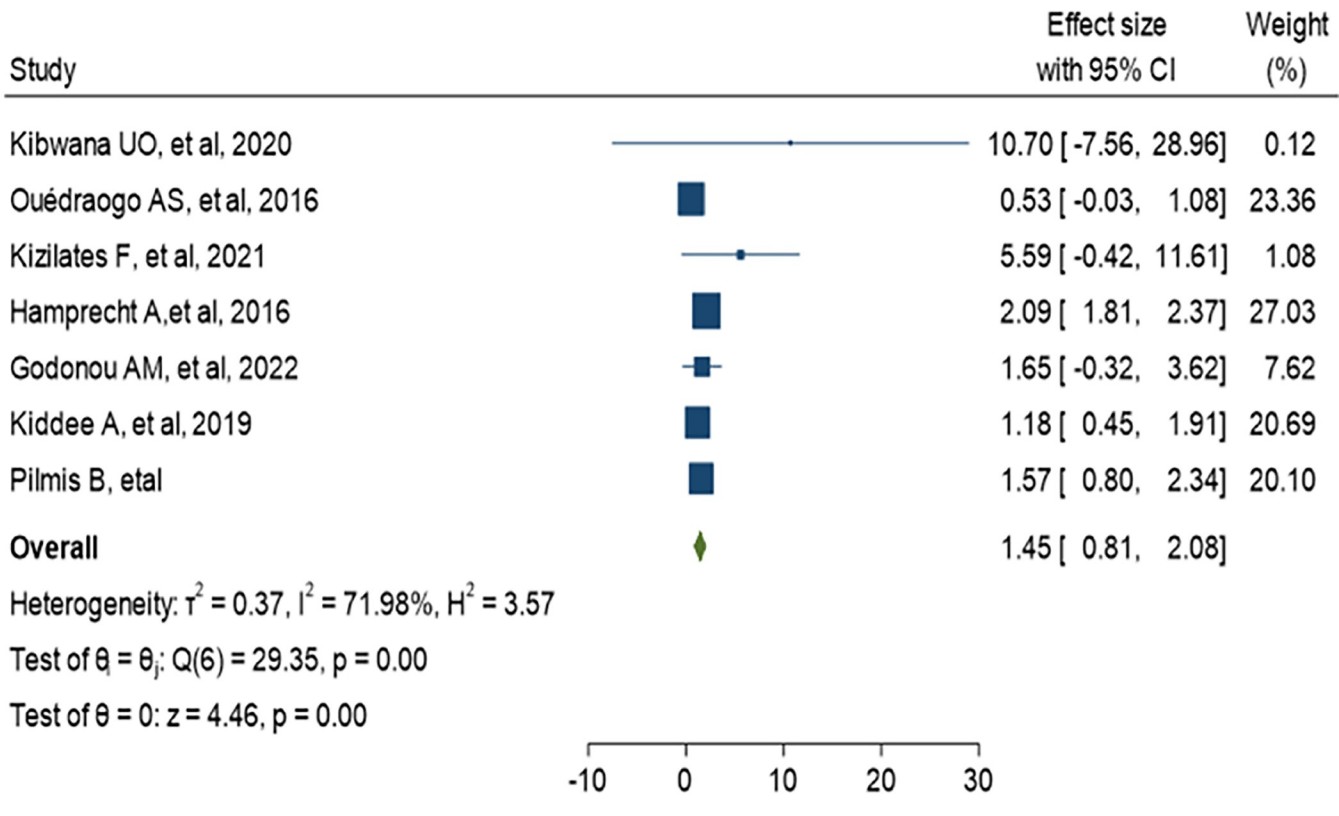

**Fig 17. Odds ratio for ESBL colonization and previous antibiotics.**

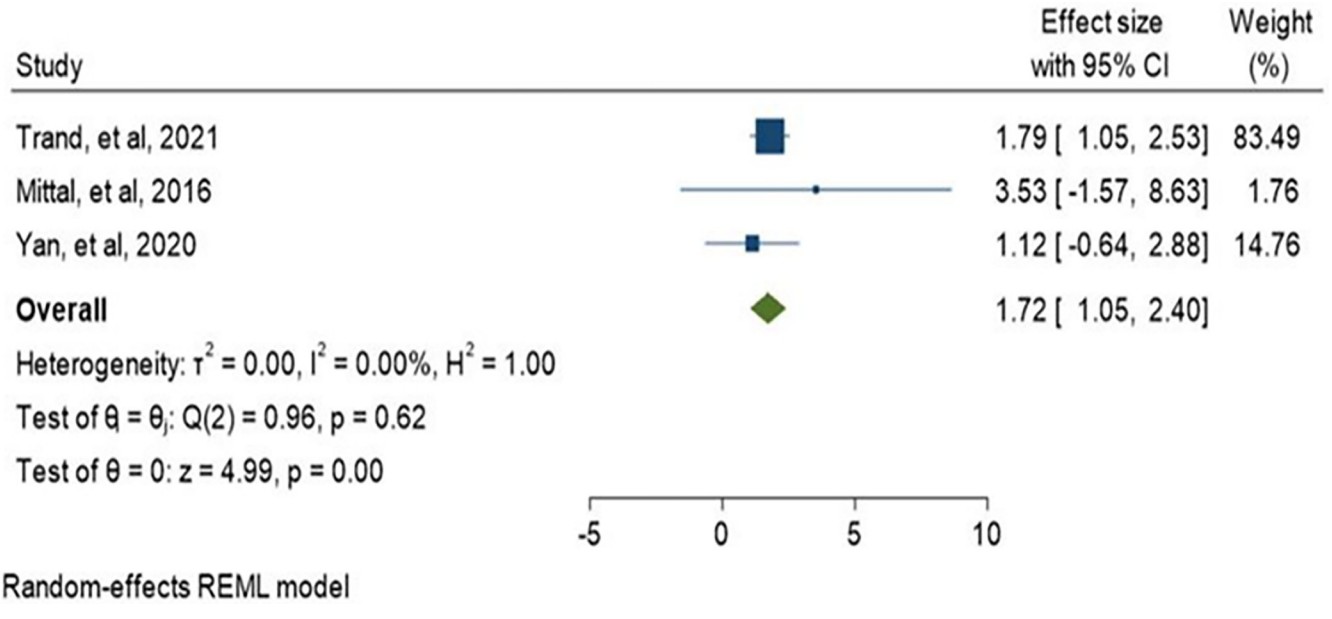

**Fig 18. Forest plot for carbapenem and CRE colonization.**

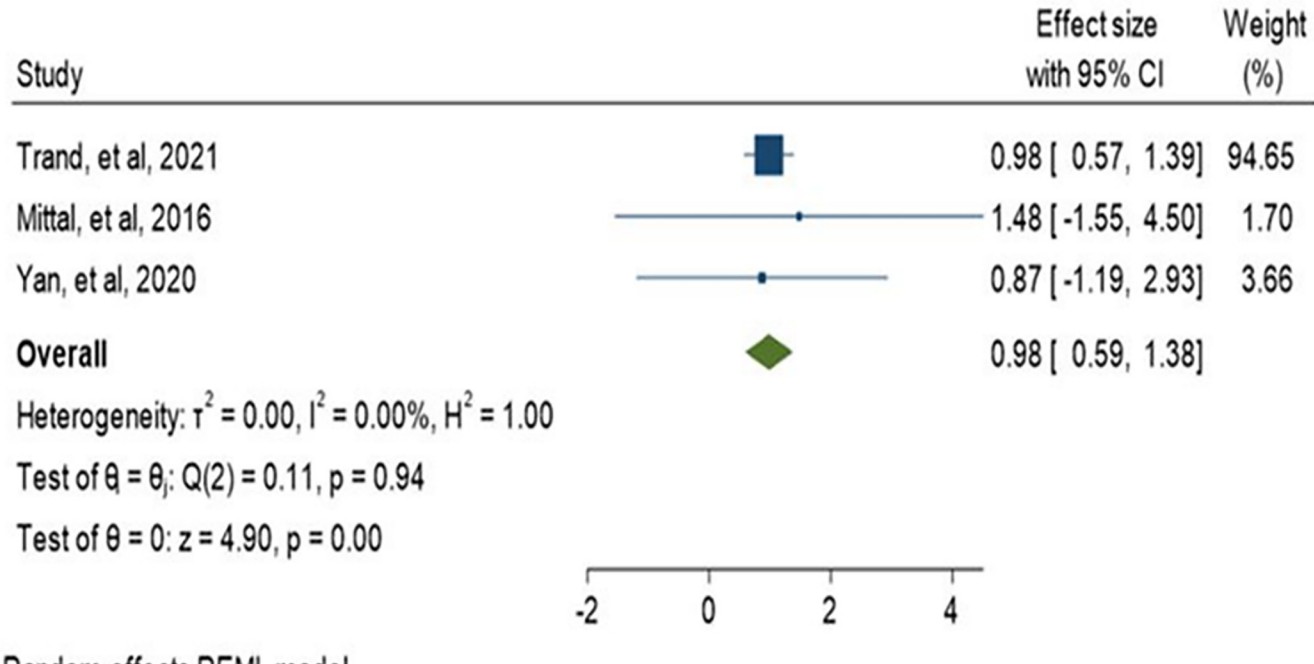

**Fig 19. Forest plot for fluoroquinolone and CRE colonization.**

**Table 4. Meta-analysis of association between comorbidities and ESBL-PE /CRE colonization.**

| Comorbidities | Included studies | Heterogeneity (I²) | P-value | OR | 95% CI |
|---|---|---|---|---|---|
| Chronic diseases | 2 | 0.00% | <0.001 | 2.02 | 1.17–2.87 |
| Surgery | 2 | 0.00% | 0.01 | 0.70 | 0.21–1.18 |
| Renal disease | 3 | 49.56% | 0.04 | 1.78 | 0.06–3.50 |
| Diabetics | 2 | 0.00% | 0.02 | 1.42 | 0.20–2.65 |
| Any catheterization | 3 | 0.00% | <0.001 | 1.04 | 0.99–1.09 |
| At admission | 2 | 99.54% | <0.001 | 14.77 | -1.35–30.90 |
| After admission | 2 | 99.90% | <0.001 | 45.63 | 0.86–92.12 |

OR = Odds ratio, CI = Confidence interval.

## 6. Conclusion

The overall colonization rate of ESBL-PE and CRE were high. Risk factors such as previous antibiotic treatment and the duration in addition to a long stay in the hospital stay were associated with ESBL-PE and CRE colonization. In addition, comorbidities such as renal diseases and diabetic diseases were linked with the colonization of ESBL-PE and/or CRE.

## Supporting information

**S1 Checklist. PRISMA 2020 checklist.**
(TIF)

## Acknowledgments

We acknowledge the authors of each study. We wish to thank Penn IIZD International Mentoring Initiative for networking opportunities.

## Author Contributions

**Conceptualization:** Dessie Abera, Abel Abera Negash, Ken Cadwell.

**Data curation:** Dessie Abera, Ayinalem Alemu.

**Formal analysis:** Dessie Abera, Ayinalem Alemu.

**Investigation:** Ayinalem Alemu, Abel Abera Negash.

**Methodology:** Dessie Abera, Ayinalem Alemu.

**Supervision:** Adane Mihret, Abel Abera Negash, Woldaregay Erku Abegaz, Ken Cadwell.

**Validation:** Adane Mihret, Abel Abera Negash, Woldaregay Erku Abegaz, Ken Cadwell.

**Writing – original draft:** Dessie Abera.

**Writing – review & editing:** Dessie Abera, Ayinalem Alemu, Adane Mihret, Abel Abera Negash, Woldaregay Erku Abegaz, Ken Cadwell.

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
