## [Decision Letter · Decision Letter 0]

12 May 2023

PONE-D-23-02816Colonization with exteneded spectrum beta-lactamase and Carbapenem producing Enterobacteriacae among hospitalized patients at the global level: a systematic review and meta-analysis.PLOS ONE

Dear Dr. Aefera,

Thank you for submitting your manuscript to PLOS ONE. After careful consideration, we feel that it has merit but does not fully meet PLOS ONE’s publication criteria as it currently stands. Therefore, we invite you to submit a revised version of the manuscript that addresses the points raised during the review process.

We look forward to receiving your revised manuscript.

Kind regards,

Nabi Jomehzadeh, Ph.D (Assistant Professor)

Academic Editor

PLOS ONE

Journal Requirements:

Additional Editor Comments (if provided):

We have received the reports from our advisors on your manuscript, "Colonization with exteneded spectrum beta-lactamase and Carbapenem producing Enterobacteriacae among hospitalized patients at the global level: a systematic review and meta-analysis," submitted to "PloS One."

Based on the advice received, I feel your manuscript could be reconsidered for publication should you be prepared to incorporate changes, as suggested by reviewers.

Reviewers' comments:

Reviewer's Responses to Questions

**Comments to the Author**

1. Is the manuscript technically sound, and do the data support the conclusions?

Reviewer #1: Yes

Reviewer #2: Partly

2. Has the statistical analysis been performed appropriately and rigorously? 

Reviewer #1: Yes

Reviewer #2: Yes

3. Have the authors made all data underlying the findings in their manuscript fully available?

Reviewer #1: Yes

Reviewer #2: Yes

4. Is the manuscript presented in an intelligible fashion and written in standard English?

Reviewer #1: Yes

Reviewer #2: Yes

5. Review Comments to the Author

Reviewer #1: -The Line 2. To replace the carbapenem with carbapenemases producing Enterobacteriaceae in the title to be more organized

-In line 25, mention carbapenemases instead of carbapenem to describe the study enzymes uniformly.

-In line 47, CRE is deleted and replaced by MDRB

-In line 75, The following references must be used to support the idea:-

Al-Qaysi, A. K., Al-Ouqaili, . M. T. & Al-Meani, S. A. (2020). Ciprofloxacin- and gentamicin-mediated inhibition of Pseudomonas aeruginosa biofilms is enhanced when combined with the volatile oil from Eucalyptus camaldulensis. SRP, 11 (7), 98-105.

In line 135, I think not to write in the world but most of the world where there are studies from the countries which are not cited. Thus, we will provide you with some important and so related works to create a study more comprehensive

-To table 2, Please use the following references and refer to the works in the following table and the country. This is important for your study to be more comprehensive and informative:-

1-AL-KUBAISY SH, HUSSEIN, RA, AL-OUQAILI, MTS. (2020). Molecular Screening of Ambler class C and extended spectrum β-lactamases in multi-drug resistant Pseudomonas aeruginosa and selected species of Enterobacteriaceae. International Journal of Pharmaceutical Research | Jul - Sep 2020 | Vol 12 | Issue 3.

2-Al-Ouqaili, MTS, Khalaf EA, Al-Kubaisy SH. (2020). DNA Sequence Analysis of BlaVEB Gene Encoding Multi-drug Resistant and Extended-spectrum β-lactamases Producer Isolates of Enterobacteriaceae and Pseudomonas aeruginosa. The Open Microbiology Journal, 4: 40-47.

3-Al-Ouqaili, MTS, Al-Taei, SA, Al-Najjar A. Molecular Detection of Medically Important Carbapenemases Genes Expressed by Metallo-β-lactamase Producer Isolates of Pseudomonas aeruginosa and Klebsiella pneumoniae. Asian Journal of Pharmaceutics • Jul -Sep 2018 (Suppl ) • 12 (3) | S991.

4-Khalaf, EA, Al-Ouqaili, MTS. Molecular detection and sequencing of SHV gene encoding for extended-spectrum β-lactamases produced by multidrug resistance some of the Gram-negative bacteria. International Journal of Green Pharmacy • Oct-Dec 2018 (Suppl) • 12 (4) | S910-S918.

-In line 272, It is important to refer to the following reference: Al-Ouqaili, MTS., Jal'oot, AS., Badawy, AS. Identification of an OprD and bla(IMP) Gene-mediated Carbapenem Resistance in Acinetobacter baumannii and Pseudomonas aeruginosa among Patients with Wound Infections in Iraq. Asian Journal of Pharmaceutics., volume 12, Issue 3, 2019 Page S959-S965-

-In the conclusion statement, particularly, line 388, Please do add to line 388 as follow: - antibiotic treatment and the duration in addition to a long stay in the hospital

Reviewer #2: It is recommended to enter three valid databases that are under review. At least Embase or Scopus should be added.

Which method did you use to categorize the I2 values? Please state the reference

Why didn't you use the Egger and Begg tests for assessing publication bias?

(209): Please report your findings using subgroup analysis on the type of study (cohort, case control or cross sectional)

6. PLOS authors have the option to publish the peer review history of their article (what does this mean?). If published, this will include your full peer review and any attached files.

Reviewer #1: No

Reviewer #2: **Yes: **OK

---

## [Author Response · Author response to Decision Letter 0]

21 Jul 2023

We thank all reviewers and the editor.

---

## [Decision Letter · Decision Letter 1]

13 Sep 2023

PONE-D-23-02816R1

Colonization with exteneded spectrum beta-lactamase and carbapenemases producing Enterobacteriacae  among hospitalized patients at the global level: a systematic review and meta-analysis .

PLOS ONE

Dear Dr. Aefera,

Thank you for submitting your manuscript to PLOS ONE. After careful consideration, we feel that it has merit but does not fully meet PLOS ONE’s publication criteria as it currently stands. Therefore, we invite you to submit a revised version of the manuscript that addresses the points raised during the review process.

We look forward to receiving your revised manuscript.

Kind regards,

Nabi Jomehzadeh, Ph.D (Assistant Professor)

Academic Editor

PLOS ONE

Journal Requirements:

Additional Editor Comments :

Comments from PLOS Editorial Office: We note that one or more reviewers has recommended that you cite specific previously published works. As always, we recommend that you please review and evaluate the requested works to determine whether they are relevant and should be cited. It is not a requirement to cite these works. We appreciate your attention to this request.

Reviewers' comments:

Reviewer's Responses to Questions

**Comments to the Author**

1. If the authors have adequately addressed your comments raised in a previous round of review and you feel that this manuscript is now acceptable for publication, you may indicate that here to bypass the “Comments to the Author” section, enter your conflict of interest statement in the “Confidential to Editor” section, and submit your "Accept" recommendation.

Reviewer #3: All comments have been addressed

2. Is the manuscript technically sound, and do the data support the conclusions?

Reviewer #3: Yes

3. Has the statistical analysis been performed appropriately and rigorously? 

Reviewer #3: Yes

4. Have the authors made all data underlying the findings in their manuscript fully available?

Reviewer #3: Yes

5. Is the manuscript presented in an intelligible fashion and written in standard English?

Reviewer #3: Yes

6. Review Comments to the Author

Reviewer #3: I have attached comments for consideration.

Line 1: Colonization with exteneded spectrum beta-lactamase and carbapenemases producing Enterobacteriacae among hospitalized patients at the global level: a systematic review and meta-analysis –in the title, please double check the spelling of the word extended

7. PLOS authors have the option to publish the peer review history of their article (what does this mean?). If published, this will include your full peer review and any attached files.

Reviewer #3: **Yes: **Gerald Mboowa

---

## [Author Response · Author response to Decision Letter 1]

27 Sep 2023

We thank you all reviewers and editor comments.

---

## [Decision Letter · Decision Letter 2]

16 Oct 2023

Colonization with extended spectrum beta-lactamase and carbapenemases producing Enterobacteriaceae  among hospitalized patients at the global level: a systematic review and meta-analysis.

PONE-D-23-02816R2

Dear Dr. Aefera,

We’re pleased to inform you that your manuscript has been judged scientifically suitable for publication and will be formally accepted for publication once it meets all outstanding technical requirements.

Kind regards,

Nabi Jomehzadeh, Ph.D (Assistant Professor)

Academic Editor

PLOS ONE
---

## [Editor Report · Acceptance letter]

14 Nov 2023

PONE-D-23-02816R2 

Colonization with extended spectrum beta-lactamase and carbapenemases producing *Enterobacteriaceae*  among hospitalized patients at the global level: a systematic review and meta-analysis. 

Dear Dr. Abera:

I'm pleased to inform you that your manuscript has been deemed suitable for publication in PLOS ONE. Congratulations! Your manuscript is now with our production department. 

Kind regards, 

on behalf of

Dr. Nabi Jomehzadeh 

Academic Editor

PLOS ONE